# Structural insights into the GTP-driven monomerization and activation of a bacterial LRRK2 homolog using allosteric nanobodies

**Christian Galicia[1,2], Giambattista Guaitoli[3,4], Marcus Fislage[1,2], Christian Johannes Gloeckner[3,4], Wim Versées[1,2]***

[1]Structural Biology Brussels, Vrije Universiteit Brussel, Brussels, Belgium; [2]VIB-VUB Center for Structural Biology, VIB, Brussels, Belgium; [3]German Center for Neurodegenerative Diseases, Tübingen, Germany; [4]Institute for Ophthalmic Research, Center for Ophthalmology, University of Tübingen, Tübingen, Germany

**\*For correspondence:**
wim.versees@vub.be

**Competing interest:** The authors declare that no competing interests exist.

**Abstract** Roco proteins entered the limelight after mutations in human LRRK2 were identified as a major cause of familial Parkinson's disease. LRRK2 is a large and complex protein combining a GTPase and protein kinase activity, and disease mutations increase the kinase activity, while presumably decreasing the GTPase activity. Although a cross-communication between both catalytic activities has been suggested, the underlying mechanisms and the regulatory role of the GTPase domain remain unknown. Several structures of LRRK2 have been reported, but structures of Roco proteins in their activated GTP-bound state are lacking. Here, we use single-particle cryo-electron microscopy to solve the structure of a bacterial Roco protein (CtRoco) in its GTP-bound state, aided by two conformation-specific nanobodies: Nb_{Roco1} and Nb_{Roco2}. This structure presents CtRoco in an active monomeric state, featuring a very large GTP-induced conformational change using the LRR-Roc linker as a hinge. Furthermore, this structure shows how Nb_{Roco1} and Nb_{Roco2} collaborate to activate CtRoco in an allosteric way. Altogether, our data provide important new insights into the activation mechanism of Roco proteins, with relevance to LRRK2 regulation, and suggest new routes for the allosteric modulation of their GTPase activity.

## eLife assessment

The **fundamental** study by Galicia et al. captured the GTP-bound active structure of CtRoco, a homolog of human LRRK2, using conformation-specific nanobodies. This **convincing** body of work reports the first structure of a GTP-bound ROCO protein, illustrating how GTP facilitates the dimer-to-monomer transition of CtRoco and functional activation.

## Introduction

The Roco proteins are a family of large multidomain GTPases, with representatives in all domains of life (*Bosgraaf and Van Haastert, 2003*). The minimal domain arrangement that characterizes this family consists of a Roc (Ras of complex proteins) domain bearing the GTPase activity, followed by a COR (C-terminal of Roc) domain that can be further subdivided into a CORA and a CORB domain (*Wauters et al., 2019*). This protein family has attracted particular interest since the discovery that mutations in one of the four human Roco representatives, called leucine-rich repeat kinase 2 (LRRK2), are one of the major causes of familial Parkinson's disease (PD) (*Paisán-Ruíz et al., 2004*; *Zimprich*

*et al., 2004*). Additionally, genome-wide association studies have also linked this gene to idiopathic PD, as well as to some other diseases such as Crohn's disease (*Di Maio et al., 2018*; *Gilks et al., 2005*; *Hui et al., 2018*). LRRK2 is a very large and complex protein, where the central Roc-COR domains are preceded by armadillo (ARM), ankyrin (ANK), and leucine-rich repeat (LRR) domains, and followed by a Ser/Thr protein kinase and a WD40 domain. Several Rab proteins have been identified as physiological substrates of the LRRK2 kinase domain (*Liu et al., 2018*; *Steger et al., 2016*; *Steger et al., 2017*).

PD-associated mutations in LRRK2 are concentrated in the catalytic Roc-COR and kinase domains, and commonly lead to an increase in kinase activity and – depending on the experimental set-up – a decrease in GTPase activity (*Kalogeropulou et al., 2022*; *Lewis et al., 2007*; *West et al., 2005*). Moreover, recent data also suggest a reciprocal cross-talk between both catalytic activities (*Gilsbach et al., 2023*; *Störmer et al., 2023*; *Weng et al., 2023*). Nevertheless, while the therapeutic targeting of the LRRK2 kinase domain has been a major focus of research in the last two decades, the regulation and targeting of the Roc-COR domains is only recently gaining more attention (*Cogo et al., 2022*; *Helton et al., 2021*; *Park et al., 2022*; *Pathak et al., 2023*).

The first important structural insights into the functioning of the Roc-COR domains and the associated GTPase activity came from work with a Roco protein from the bacterium *Chlorobaculum tepidum* (CtRoco) (*Deyaert et al., 2019*; *Gotthardt et al., 2008*). Bacterial Roco proteins are simpler, only consisting of the core LRR-Roc-COR domains, and hence lack the kinase domain. A crystal structure of CtRoco in a nucleotide-free state agrees with earlier biophysical data, showing that CtRoco is a homodimer in the nucleotide-free state, while it monomerizes upon GTP binding and exists in a concentration-dependent monomer–dimer equilibrium in the GDP-bound state (*Deyaert et al., 2019*; *Deyaert et al., 2017b*). A PD analogous mutation in the Roc domain of CtRoco (L487A) shifts this equilibrium toward the dimer and decreases the GTPase activity, suggesting that monomerization is a requirement for normal GTPase activity.

Recently, several cryo-electron microscopy (cryo-EM) structures of LRRK2 have provided crucial new insights into the LRRK2 function and regulation (*Deniston et al., 2020*; *Myasnikov et al., 2021*). Full-length LRRK2 (FL-LRRK2) appears as a mixture of monomers and homodimers under the experimental conditions used, allowing Myasnikov and colleagues to solve cryo-EM structures of both oligomeric states with the Roc domain bound to GDP (*Myasnikov et al., 2021*). Interestingly, the symmetrical homodimers are formed through a CORB-CORB interface, very similar to the one found in the CtRoco homodimer. The kinase domain is present in an inactive ('DYG out') conformation, with the LRR domain wrapping around the kinase ATP-binding cleft, thereby blocking the entry of globular substrates and providing a mechanism of auto-inhibition. A recently published manuscript reports the structure of FL-LRRK2 bound to Rab29 (*Zhu et al., 2023*). In addition to kinase-inactive monomers and dimers, an intriguing third form is observed in this study where LRRK2 adopts an asymmetric dimer of dimers. Each of the constituting dimers contains one protomer displaying the kinase in an inactive form and one in an active form. In the kinase-active protomer, the N-terminal (ARM-ANK-LRR) domains become flexible, thus releasing the auto-inhibition provided through the LRR domain. These data thus put forward the LRR domain as an important regulatory element. Nevertheless, while we now have access to structures of full-length Roco proteins in the nucleotide-free form (CtRoco) and bound to GDP (FL-LRRK2), no structural data is available for any Roco protein bound to GTP, leaving the role of the Roc domain and of GTP binding in the regulation of Roco proteins enigmatic.

Nanobodies (Nbs), the single-domain fragments derived from camelid heavy-chain antibodies, form excellent tools to allosterically modify the activity of proteins and enzymes, including LRRK2 (*Singh et al., 2022*; *Uchański et al., 2020*). Previously, we generated two Nbs (Nb$_{Roco1}$ and Nb$_{Roco2}$) that bind preferentially to the monomeric GTP-bound form of CtRoco (*Leemans et al., 2020*). Correspondingly, binding of Nb$_{Roco1}$ shifts the dimer–monomer equilibrium to the monomeric state and thereby reverts the decrease in GTPase activity caused by the L487A PD-analogous mutation. However, the mechanism underlying this allosteric modulation provided by Nb$_{Roco1}$ and Nb$_{Roco2}$ is not yet understood.

Here we use single-particle cryo-EM to solve the structure of full-length CtRoco in its GTP state, stabilized by the binding of Nb$_{Roco1}$ and Nb$_{Roco2}$. This structure presents CtRoco in an active monomeric state, featuring very significant conformational changes compared to the dimer structure. These results provide important new insights into the activation mechanism and regulation of Roco proteins, and present potential routes for the allosteric modulation of their GTPase activity.

# Results

## Cryo-EM structures of nanobody-stabilized CtRoco in the GTP state reveal a monomeric 'open' conformation

While we previously showed that CtRoco undergoes a dimer to monomer transition during its GTPase cycle (*Deyaert et al., 2017b*; *Figure 1A*), so far only an X-ray crystal structure of the nucleotide-free dimeric state of the protein is available. This structure shows CtRoco as a rather compact dimer where the LRR domains fold back on the Roc-COR domains (*Figure 1B*; *Deyaert et al., 2019*). Obtaining well-diffracting crystals of the GTP-bound CtRoco protein is most likely hampered by the high internal flexibility of the protein, as suggested by SAXS experiments (*Deyaert et al., 2017b*). To reduce this flexibility and stabilize the protein in the monomeric GTP-bound state, we reasoned we could make use of two conformation-specific Nbs (Nb$_{Roco1}$ and Nb$_{Roco2}$) that preferentially bind the CtRoco GTP state (*Leemans et al., 2020*).

For cryo-EM sample and grid preparation, the nucleotide-free full-length CtRoco protein was incubated with an excess of the non-hydrolyzable GTP analog GTPγS and with either Nb$_{Roco1}$ (CtRoco-Nb$_{Roco1}$) or with both Nb$_{Roco1}$ and Nb$_{Roco2}$ (CtRoco-Nb$_{Roco1}$-Nb$_{Roco2}$), and the complexes were purified using size-exclusion chromatography (SEC). Cryo-EM data was collected for both complexes and map reconstructions of CtRoco-Nb$_{Roco1}$ and CtRoco-Nb$_{Roco1}$-Nb$_{Roco2}$ were refined to 8.3 Å and 7.7 Å, respectively (*Figure 1C*, *Figure 1—figure supplements 1 and 2*, *Table 1*). Both resulting maps clearly show CtRoco in a monomeric and elongated 'open' conformation, in contrast to the more globular dimeric 'closed' conformation previously observed in the nucleotide-free CtRoco crystal structure. Difficulties in aligning particles during data processing reflect the high flexibility of the N-terminal LRR domain vis-à-vis the C-terminal domains. This can be observed in the 2D classes from both data sets: while some display the full particle, most classes center on one half of the protein with the other half fading away as a result of flexibility. The majority of the 2D classes center on the N-terminal LRR domain and a few on the C-terminal Roc-COR domains (*Figure 1—figure supplement 3*). Similarly, 3D classification of the CtRoco-Nb$_{Roco1}$-Nb$_{Roco2}$ data set produced only one class representing a nearly full reconstruction, but multiple classes centering on the LRR domain with deficient density for the rest of the protein (*Figure 1—figure supplement 1*). Further refinement with particles that converged in visible density for both domains resulted in the rather low-resolution map reconstructions for the entire protein. To avoid these issues linked to inter-domain flexibility and increase the quality of the local reconstruction of each domain, we created focused maps from the CtRoco-Nb$_{Roco1}$-Nb$_{Roco2}$ data set by processing separately the N-terminal and the C-terminal centered classes, which produced map reconstructions with mean resolutions of 3.6 Å and 3.9 Å, respectively (*Figure 1C*, *Figure 1—figure supplement 1*, *Table 1*). These focused maps were used for all further interpretation and model building, and were finally fitted on the 7.7 Å map reconstruction of the (nearly) full protein reconstruction to create a composite map showing the relative orientation of these domains to each other (*Figure 1C and D*). This resulted in a good-quality interpretable map for the LRR domain bound to Nb$_{Roco2}$, with unambiguous density for most amino acid side chains on the LRR-Nb$_{Roco2}$ interface (*Figure 1—figure supplement 4A*). No density can be assigned to a peptide region linking the LRR domain to the Roc domain, in agreement with the flexible nature of this linker. The quality of the map for the C-terminal part of the protein is more variable. For the Roc domain, unambiguous density allowing side chain interpretation is observed for some parts of the domain, including parts of the region that interacts with Nb$_{Roco1}$. However, while the density for GTPγS and for a large part of the Switch 2 loop could be well interpreted (*Figure 1—figure supplement 4B*), the density is ambiguous in some regions around the GTP-binding pocket and absent for most of the Switch 1 loop. The map also allows us to unambiguously place Nb$_{Roco1}$, although it is generally of lower quality in this region. In particular, the CDR loops of Nb$_{Roco1}$ could be identified but hardly any side chains could be assigned (*Figure 1—figure supplement 4C*). Also for the CORA and CORB domains, the quality of the map is variably spread. No density is observed for C-terminal residues of CORB (a.a. 892–940), which are involved in CtRoco dimerization. This is in very good agreement with previous hydrogen-deuterium exchange (HDX) experiment showing that this region becomes highly flexible upon GTP-driven monomerization (*Deyaert et al., 2019*). Nevertheless, some weak density for this region can be observed in the full reconstruction map (*Figure 1—figure supplement 1*). Finally, no density is observed for the last 150 residues of CtRoco, although in the CtRoco-Nb$_{Roco1}$ map some additional density is observed that can account for the C-terminal region. This region is probably highly flexible and also needed

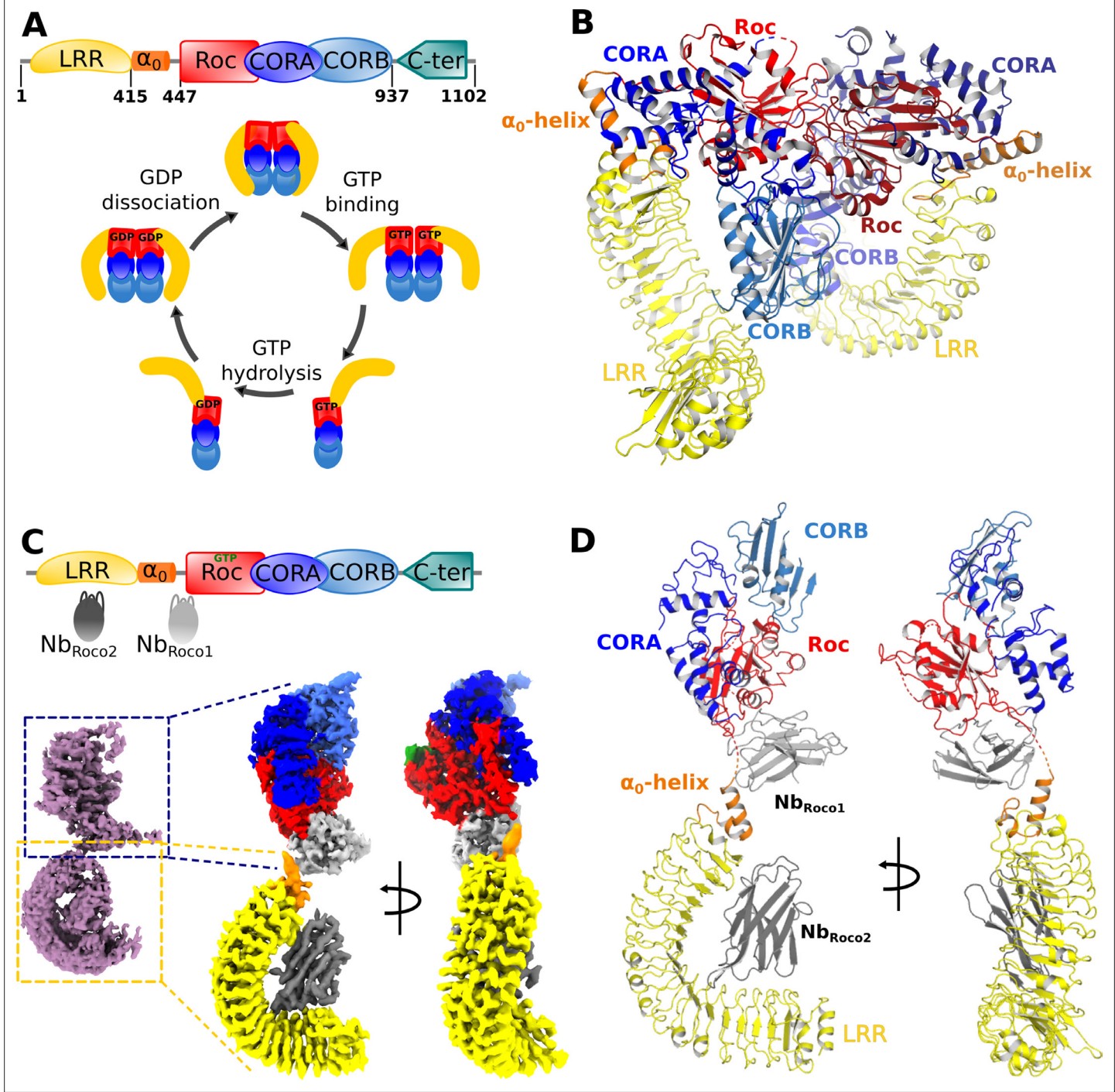

**Figure 1.** Nucleotide-induced conformational changes in CtRoco. (**A**) Schematic representation of the domain arrangement of CtRoco (top) and its anticipated conformational cycle linked to GTP binding and hydrolysis (bottom) (**Deyaert et al., 2017b**). (**B**) The previously published crystal structure of nucleotide-free CtRoco showing the protein as a compact homodimer, where the LRR domains fold back on the Roc-COR domains (PDB: 6hlu) (**Deyaert et al., 2019**). (**C**) Nb$_{Roco1}$ and Nb$_{Roco2}$ specifically bind to the CtRoco monomer in its 'GTP state' (top) (**Leemans et al., 2020**). The cryo-electron microscopy (cryo-EM) map of GTPγS-bound full-length CtRoco in complex with Nb$_{Roco1}$ and Nb$_{Roco2}$ at 7.7 Å resolution shows an elongated monomeric arrangement (bottom left). A composite map, obtained by superposing the individual cryo-EM maps focused on the C-terminal (3.9 Å) and N-terminal (3.6 Å) part of the protein, was generated (bottom right). This map is colored according to the corresponding domains and Nbs. (**D**) Final model of CtRoco-Nb$_{Roco1}$-Nb$_{Roco2}$ based on the maps shown in (**C**).

The online version of this article includes the following figure supplement(s) for figure 1:

**Figure supplement 1.** Cryo-electron microscopy (cryo-EM) workflow to obtain the maps corresponding to CtRoco-Nb$_{Roco1}$-Nb$_{Roco2}$.

*Figure 1 continued on next page*

**Figure supplement 3.** 2D classification of CtRoco-Nb$_{Roco1}$-Nb$_{Roco2}$ (representative classes).

**Figure supplement 4.** Quality of the cryo-electron microscopy (cryo-EM) map of CtRoco-Nb$_{Roco1}$-Nb$_{Roco2}$.

**Figure supplement 5.** Fourier shell correlation (FSC) plots of refined atomic models against the focused cryo-electron microscopy (cryo-EM) density maps of CtRoco-Nb$_{Roco1}$-Nb$_{Roco2}$.

to be removed to obtain diffracting crystals of the nucleotide-free CtRoco dimer (*Deyaert et al., 2017a*). Taken together, the GTPγS-bound CtRoco-Nb$_{Roco1}$-Nb$_{Roco2}$ structure clearly shows the protein in a monomeric and elongated 'open' conformation, allowing us to interpret the GTP-driven conformational changes, as well as the binding sites and the mechanism of allosteric activation of Nb$_{Roco1}$ and Nb$_{Roco2}$ (*Figure 1D*). Only data from CtRoco-Nb$_{Roco1}$-Nb$_{Roco2}$ were used for detailed analyses and will be discussed further.

## GTP-mediated 'activation' of CtRoco induces large-scale conformational changes linked to monomerization

A comparison of the current structure of monomeric GTPγS-bound CtRoco-Nb$_{Roco1}$-Nb$_{Roco2}$ with the previously solved structure of dimeric CtRoco in the nucleotide-free (NF) state (*Deyaert et al., 2019*) allows us to dissect in detail the conformational changes associated with CtRoco monomerization and activation. To achieve this, protomers of each structure were superposed using their Roc domains as a reference. The most prominent conformational change concerns the position of the LRR domain. In the GTPγS-bound structure, the LRR rotates away from the Roc-COR domains by about 135° (*Figure 2A*). The hinge point for this large movement is the linker region between the LRR and Roc domains, and in particular the so-called α$_0$-helix that precedes the actual Roc domain. This helix is a conserved structural element of the Roco proteins, including LRRK2. In the inactive dimeric CtRoco, the α$_0$-helix is structurally inserted between the LRR and CORA domains, forming hydrophobic and electrostatic interactions with both domains (*Figure 2—figure supplement 1*). However, upon GTPγS binding this helix detaches and becomes entirely solvent exposed, as such disrupting the interaction between the LRR and Roc-COR domains (*Figure 2B*). This observation is in good agreement with our earlier HDX data, which also suggested that the region connecting the LRR and Roc domain becomes solvent exposed in the GTP-bound monomer (*Deyaert et al., 2019*).

A comparison of the Roc domains in the nucleotide-free and GTPγS-bound states reveals a number of conformational changes. A remarkable feature of the CtRoco dimer structure was the dimer-stabilized orientation of the P-loop, which would hamper direct nucleotide binding on the dimer (*Figure 2—figure supplement 2*). Correspondingly, in the current structure the P-loop changes orientation allowing GTPγS to bind, although the EM map does not allow unambiguous placement of the entire P-loop. Also the Switch 1 loop could not be fully modeled in our structure, presumably indicating some flexibility in this region despite the presence of a GTP analog. Interestingly, the Switch 1 loop harbors the site of the PD-analogous L487A mutation that leads to a stabilization of the CtRoco dimer with a concomitant decrease in GTPase activity (*Deyaert et al., 2019*). Unfortunately, an exact interpretation of this effect of the L487A mutation is hampered by the lack of a well-resolved Switch 1 loop. Another region of conformational change regards Switch 2. The EM map for this region and for the preceding interswitch β-strand is of reasonably good quality, allowing to model a large part of the backbone of Switch 2 (including the DxxG motif) (*Figure 2C*). Compared to the position of the nucleotide-free structure, the Switch 2 shifts to a position much closer to the P-loop that is not compatible with the dimeric arrangement of nucleotide-free CtRoco since it would sterically clash with the adjacent protomer (*Figure 2D*, *Figure 2—figure supplement 2*). Hence, those conformational changes in the P-loop and Switch 2 region could form an initial trigger for nucleotide-induced monomerization. A final important observation in the Roc domain concerns the very C-terminal part of Switch 2 (residues 520–533), which, in contrast to the main part of Switch 2, could not be modeled in the GTP-bound structure, potentially due to flexibility of this region in the new position of the Switch 2. However, in the nucleotide-free dimer structure this region of Switch 2 is structured and located at the interface of the Roc domain with the LRR-Roc linker and CORA (*Figure 2—figure supplement 2*).

**Table 1.** Cryo-electron microscopy (cryo-EM) data collection, refinement, and validation statistics.

| | CtRoco-Nb$_{Roco1}$-Nb$_{Roco2}$ | CtRoco-Nb$_{Roco1}$-Nb$_{Roco2}$ (LRR focused) | CtRoco-Nb$_{Roco1}$-Nb$_{Roco2}$ (Roc-COR focused) | CtRoco-Nb$_{Roco1}$ |
|---|---|---|---|---|
| **Data collection** | | | | |
| Microscope | CryoARM300 | | | CryoARM300 |
| Voltage (keV) | 300 | | | 300 |
| Electron exposure (e$^-$/Å$^2$) | 63 | | | 63 |
| Energy filter slit width | 20 eV | | | 20 eV |
| Detector | Gatan K3 | | | Gatan K3 |
| Magnification | × 60,000 | | | × 60,000 |
| Defocus range (μm) | 1–3 | | | 1–3 |
| Pixel size (Å/pix) | 0.755 | | | 0.766 |
| Number of movies | 7489 | | | 11,718 |
| Symmetry | C1 | | | C1 |
| Final particles | 48,333 | 160,925 | 38,260 | 99,460 |
| Map mean resolution (Å) | 7.7 | 3.6 | 3.9 | 8.3 |
| **Model refinement** | | | | |
| Atoms | | 4038 | 3573 | |
| Bonds (RMSD) | | | | |
| Bond length (Å) | | 0.004 | 0.003 | |
| Bond angles (°) | | 0.749 | 0.647 | |
| Validation | | | | |
| Clash score | | 13.09 | 12.94 | |
| Rotamer outliers (%) | | 0.7 | 0.3 | |
| MolProbity score | | 2.15 | 2.18 | |
| Ramachandran plot | | | | |
| Favored (%) | | 88.57 | 89.78 | |
| Allowed (%) | | 11.25 | 9.57 | |
| Outlier (%) | | 0.18 | 0.65 | |
| Mean B-factors | | | | |
| Protein | | 45.93 | 37.13 | |
| Ligand | | | 39.20 | |

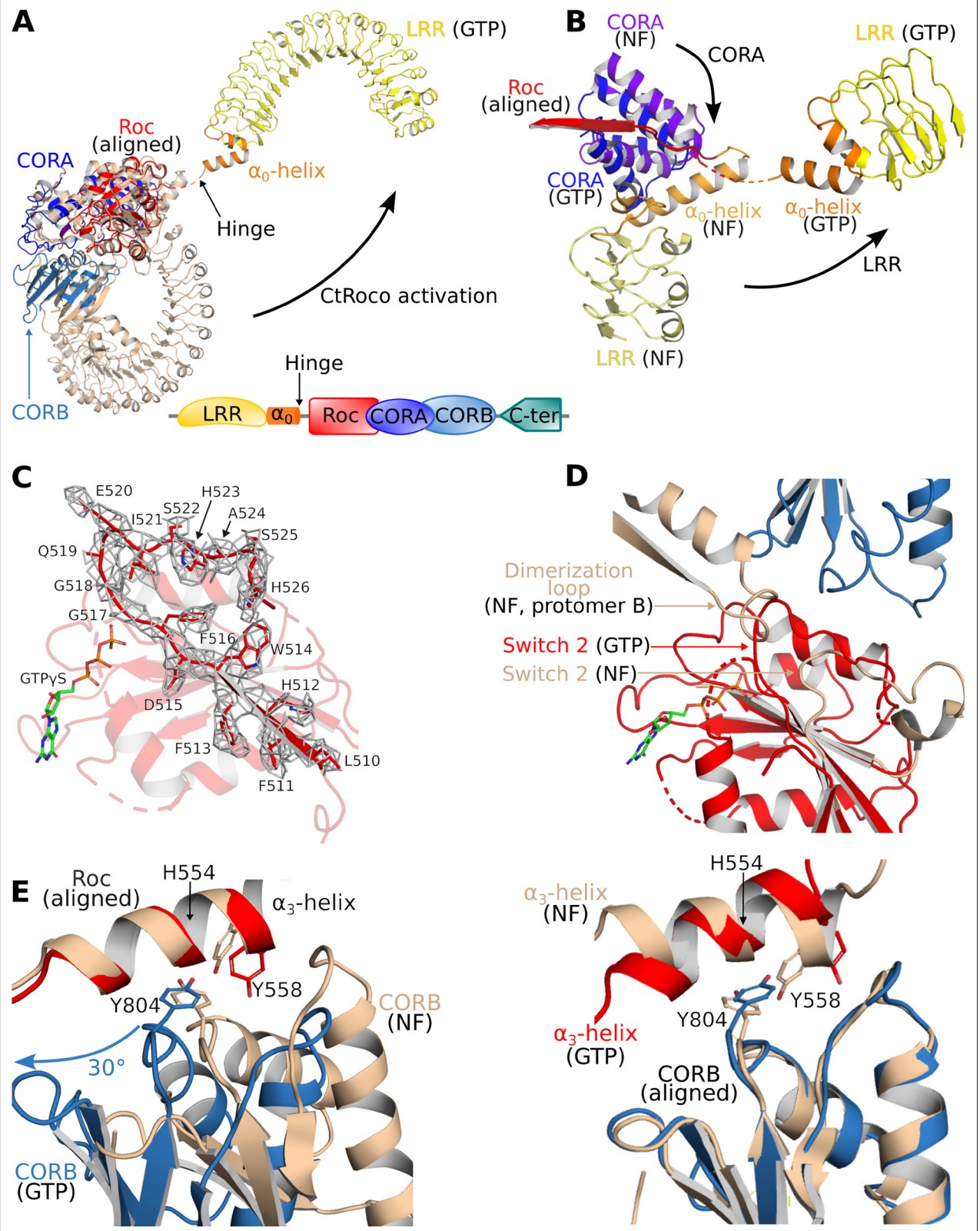

**Figure 2.** Structural changes occurring upon GTP-driven CtRoco activation. (**A**) Superposition of a subunit of the nucleotide-free CtRoco dimer (labeled 'NF'; beige) and the GTPγS-bound monomeric CtRoco-Nb$_{Roco1}$-Nb$_{Roco2}$ (labeled 'GTP', colored by domain). The Roc domains of both structures were used for the superposition. Nbs were removed in the representation of CtRoco-Nb$_{Roco1}$-Nb$_{Roco2}$ for clarity. The conformational change occurring in the LRR domain and α$_0$-helix is indicated by an arrow. (**B**) Close-up view of the displacement of the LRR, α$_0$-helix and CORA upon CtRoco activation.

*Figure 2 continued on next page*

*Figure 2 continued*

The position of CORA in the 'GTP' conformation is incompatible with the position of the $\alpha_0$-helix in the nucleotide-free conformation, providing a mechanism to relay conformational changes from Roc and CORA toward the $\alpha_0$-helix and LRR domain. The LRR, $\alpha_0$, and CORA of nucleotide-free and GTPγS-bound CtRoco are colored in different shades of yellow, orange, and blue, respectively. (**C**) Mesh representation of the cryo-electron microscopy (cryo-EM) map around the Switch 2 region and the preceding β-strand in the GTPγS-bound Roc domain. (**D**) Superposition of Roc domains and surrounding regions of nucleotide-free (beige) and GTPγS-bound CtRoco (colored according to domain). The conformation of the Switch 2 region in GTPγS-bound CtRoco would sterically clash with the 'dimerization loop' of the adjacent protomer of the nucleotide-free dimer, providing an initial trigger for nucleotide-induced monomerization. (**E**) Upon activation, the CORB domain of GTPγS-bound CtRoco (blue) displays a 30° rotational movement with regard to corresponding domain of nucleotide-free CtRoco (beige). The H554-Y558-Y804 triad (corresponding to N1437-R1441-Y1699 in LRRK2), located at the Roc-CORB interface, acts as a hinge point for this rotational movement. Left: view of a superposition using the Roc domain; right: view of a superposition using the CORB domain.

The online version of this article includes the following figure supplement(s) for figure 2:

**Figure supplement 1.** The $\alpha_0$-helix is inserted between the LRR and CORA domains in the nucleotide-free CtRoco structure.

**Figure supplement 2.** Structural changes within the Roc domain and the LRR-Roc-CORA interface upon GTPγS binding.

**Figure supplement 3.** The rotational movement of the CORB domain upon GTP-driven CtRoco activation induces monomerization.

In this way, the conformational changes induced by GTPγS binding could be relayed via the Switch 2 toward the LRR and CORA domains, and vice versa.

In addition to the C-terminal dimerization part of CORB that becomes unstructured, also other large conformational changes are observed in the CORA and CORB domains of CtRoco upon GTPγS binding. The most prominent change in CORA concerns a shift of its N-terminal part (residues 626–693) toward a position that is incompatible with the position of the LRR and the Roc-LRR linker region in the CtRoco dimer structure (*Figure 2B*). In particular, the third helix of CORA (residues 666–678) would form sterical clashes with the $\alpha_0$-helix that links the LRR and Roc domains, thus linking the conformational changes in CORA to those in the $\alpha_0$-helix and LRR domain. The displacements occurring in CORB are even more pronounced. In the GTPγS-bound monomer, the N-terminal part of CORB (residues 798–891) undergoes a global rotational movement of about 30° in the direction of the dimer interface of the nucleotide-free dimer (*Figure 2E*, *Figure 2—figure supplement 3*). Such an orientation would be incompatible with the dimeric arrangement observed in the nucleotide-free state due to severe steric clashes with the Roc domain of the adjacent subunit. Interestingly, this rotational movement of CORB seems to use the H554-Y558-Y804 triad on the interface of Roc and CORB as a pivot point (*Figure 2E*). Mutation of either of the corresponding residues in LRRK2 (N1437, R1441, Y1699, respectively) is associated with PD and leads to LRRK2 activation. Residues H554 and Y558 are located on the Roc $\alpha_3$-helix, which was previously suggested to be an important element in the activation of LRRK2 (*Kalogeropulou et al., 2022*). Indeed, while the orientation of the $\alpha_3$-helix with respect to the rest of the Roc domain only undergoes small changes upon GTPγS binding, it can be observed that this helix undergoes a 'seesaw-like' movement with respect to the CORB domain. A similar rearrangement was previously also observed for Rab29-mediated activation of human LRRK2 (*Störmer et al., 2023*; *Zhu et al., 2023*).

A comparison of the GTPγS-bound CtRoco monomer with the nucleotide-free dimer thus clearly provides several intertwined pathways linking GTP binding to monomerization and large-scale intra-subunit conformational changes.

## Nb$_{Roco1}$ and Nb$_{Roco2}$ allosterically lock CtRoco in its active conformation

We previously showed that Nb$_{Roco1}$ binds CtRoco in a conformation-specific way, with the highest affinity for the GTP-bound state, while no binding was observed on the nucleotide-free CtRoco dimer. As a result, Nb$_{Roco1}$ shifts the CtRoco dimer–monomer equilibrium toward the monomeric state and increases the GTPase activity (*Leemans et al., 2020*). Albeit less pronounced, a similar specificity for the GTP-bound state of CtRoco is observed for Nb$_{Roco2}$. The current structure of GTPγS-bound CtRoco in complex with both Nb$_{Roco1}$ and Nb$_{Roco2}$ now reveals the mechanism underlying this conformational specificity.

Nb$_{Roco1}$ binds to CtRoco on the interface of the Roc and CORA domains and the hinge region connecting the LRR to the Roc domain (including the $\alpha_0$-helix). As such it is strategically positioned to 'monitor' the nucleotide-induced conformational changes occurring in CtRoco and to stabilize the active conformation (*Figures 1D and 3A*). Specifically, the CDR1 loop of Nb$_{Roco1}$ is within relatively

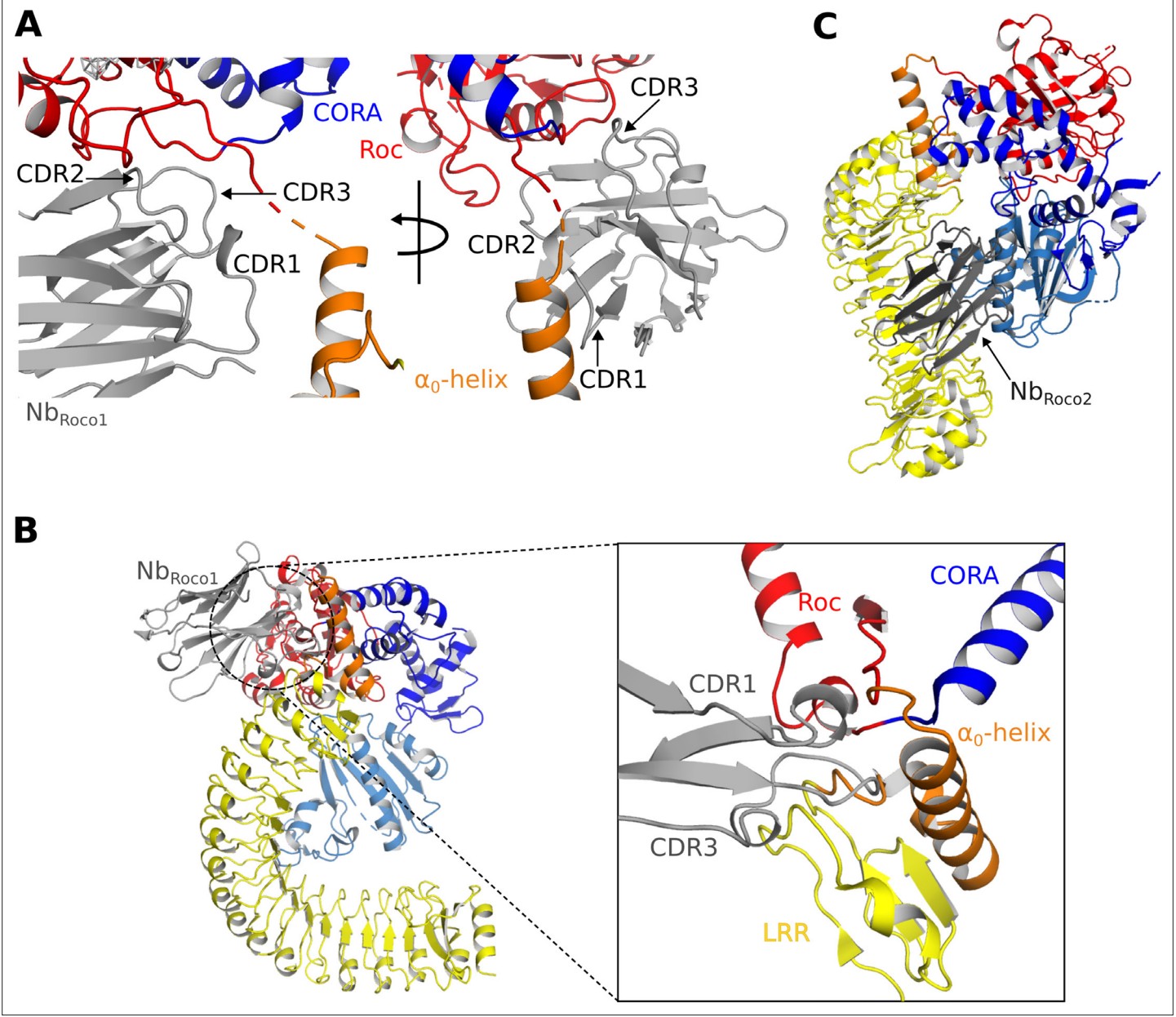

**Figure 3.** Mechanisms of the allosteric activation of CtRoco by Nb$_{Roco1}$ and Nb$_{Roco2}$. (**A**) Nb$_{Roco1}$ binds at the interface of the Roc, CORA, and LRR domains, close to the α$_0$-helix and the hinge region of the LRR movement. (**B**) Superposition of Nb$_{Roco1}$ on the conformation adopted by the nucleotide-free CtRoco shows considerable sterical clashes between the CDR1 and CDR3 regions of Nb$_{Roco1}$ and the LRR domain and α$_0$-helix of nucleotide-free CtRoco (see inset). This provides a mechanism for the Nb$_{Roco1}$-mediated stabilization of CtRoco in its GTP-bound active conformation. (**C**) Superposition of Nb$_{Roco2}$ on the conformation adopted by the nucleotide-free CtRoco shows that the position of Nb$_{Roco2}$ in the curvature of the LRR domain is not compatible with the closed conformation of CtRoco, due to severe sterical clashes with the CORB domain.

close distance to the linker region connecting the LRR domain to the Roc domain and to the α$_0$-helix, although the local resolution in this area does not allow unequivocal placement of the residue side chains (*Figure 3A*). The CDR2 loop mainly interacts with the Roc domain, including the interswitch region, while it can potentially also interact with the C-terminal end of the LRR-Roc linker. The CDR3 loop interacts with the region linking the Roc domain to CORA (*Figure 3A*). This binding mode of Nb$_{Roco1}$ would clearly be incompatible with binding to CtRoco in its inactive nucleotide-free dimeric state. Indeed, superposition of the current structure on the CtRoco dimer shows severe sterical overlap between the Nb$_{Roco1}$ CDR3 and the LRR, Roc and LRR-Roc linker of the CtRoco dimer, and, to a lesser extent, also between CDR1 and the LRR-Roc linker (*Figure 3B*). Hence, binding of Nb$_{Roco1}$

to CtRoco would be expected to shift the equilibrium toward the open active state, explaining the observed preference for the GTP-bound monomeric conformation.

In the density map, $Nb_{Roco2}$ can easily be identified and placed on the concave side of the LRR domain (*Figure 1—figure supplement 4A*). The main interactions occur between the CDR2 loop and the first two repeats of the LRR domain, and, in particular, between the CDR3 loop and the central region of the LRR (repeats 5–10). From CDR1, only K27 seems to be involved in the binding by making interactions with E243 and Q245 in repeat 11 of the LRR domain. Apart from these contributions of the CDR loops, also framework residues of $Nb_{Roco2}$ are implicated in the interaction with CtRoco, with in particular its N-terminal and C-terminal β-strands interacting with the very C-terminal repeat of the LRR. When superposing the active and inactive conformations of CtRoco, one can immediately appreciate that the placement of $Nb_{Roco2}$ is incompatible with the LRR conformation in the inactive nucleotide-free state (*Figure 3C*). In the latter state, the LRR folds back on the COR domains and interacts with the CORB through its repeats 9–11. Hence, this provides a mechanism for the conformational specificity of $Nb_{Roco2}$. Nevertheless, we previously found that $Nb_{Roco2}$ does bind to nucleotide-free CtRoco, albeit with a lower affinity compared to the GTP state (*Leemans et al., 2020*). This indicates some flexibility in the position of the LRR domain, regardless of the CtRoco nucleotide state, with $Nb_{Roco2}$ playing a more subtle role in shifting the equilibrium toward the monomeric open conformation. The latter is in agreement with our observation that the elongated open CtRoco conformation is also observed as the main species in the CtRoco-$Nb_{Roco1}$ structure in the absence of $Nb_{Roco2}$ (*Figure 1—figure supplement 2*).

## Crosslinking MS experiments confirm the Nb-binding sites and the induced conformational changes

To further complement and confirm our structural data, we subsequently used crosslinking mass spectrometry (CX-MS) to map the Nb-induced conformational changes in CtRoco, as well as the binding sites of $Nb_{Roco1}$ and $Nb_{Roco2}$. Hereto, we used the lysine-specific and CID-cleavable crosslinker disuccinimidyl sulfoxide (DSSO) (*Kao et al., 2011*). Considering the length of the DSSO spacer and the lysine side chains, the theoretical upper limit for the distance between the α carbon atoms of crosslinked lysines is ~26 Å, while also taking protein dynamics into account leads to a cutoff distance of 35 Å, thus also allowing to draw inter- and intramolecular interactions within this resolution limit (*Erzberger et al., 2014*; *Kao et al., 2011*). CX-MS was applied to GTPγS-bound CtRoco either in the absence of Nbs, or bound to $Nb_{Roco1}$, $Nb_{Roco2}$ or both Nbs. Overall, the obtained crosslinking data for the proteins in solution correspond well with the cryo-EM structures and previous biochemical data. In the GTPγS-bound CtRoco protein, in the absence of Nbs, still a considerable number of crosslinks are observed between the LRR domain and the other CtRoco domains, illustrating the dynamical nature of this protein state, where the LRR most probably samples both the 'open' and 'closed' states (*Figure 4A*). The number of crosslinks between the LRR domain and the other protein domains is strongly decreased upon binding of either of the two Nbs, indicating that they stabilize the open conformation of CtRoco (*Figure 4A–C*). This is most prominent for $Nb_{Roco1}$ in agreement with the strong conformational specificity of this Nb. Interestingly, in the presence of both $Nb_{Roco1}$ and $Nb_{Roco2}$, all the crosslinks between the LRR and C-terminal domains are lost, suggesting an additive effect of both Nbs in stabilizing the CtRoco open conformation (*Figure 4D*). Also, multiple crosslinks between the Nbs and CtRoco, as well as between both Nbs, were found. According to the cryo-EM structure, $Nb_{Roco1}$ binds to the LRR-Roc interface, and crosslinks were accordingly observed between $Nb_{Roco1}$-K68 and residue K443 in the LRR-Roc ($α_0$) linker region. $Nb_{Roco1}$-K68 also forms crosslinks with two lysines within the Roc domain (K583 and K611), and $Nb_{Roco1}$-K90 is crosslinked to K583 (*Figure 4D*, *Figure 4—figure supplement 1*). However, surprisingly, no crosslinks were observed between $Nb_{Roco2}$ and the LRR domain, while the cryo-EM structure and previous biochemical data unambiguously show binding to the LRR (*Leemans et al., 2020*). This can be explained by the compact folding of the LRR domain, which allows only very little rotational freedom for the lysine residues and causes a poor coverage by crosslinking data, as previously observed (*Guaitoli et al., 2016*).

## $Nb_{Roco1}$ and $Nb_{Roco2}$ have a synergistic effect

While $Nb_{Roco1}$ and $Nb_{Roco2}$ both bind to CtRoco in a conformation-specific way, with preference for the monomeric GTP-bound conformation, this feature is most pronounced in $Nb_{Roco1}$ for which we

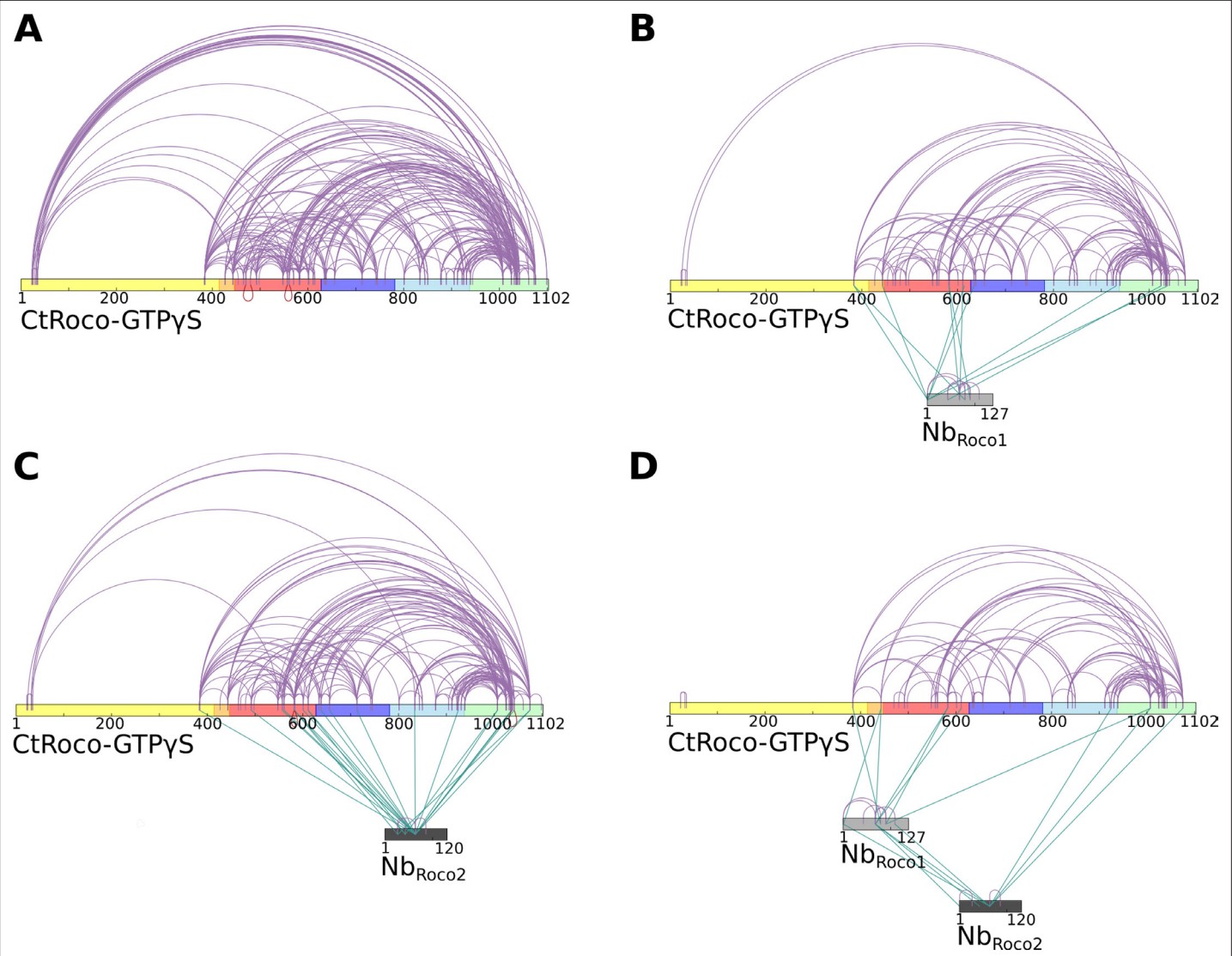

**Figure 4.** Crosslinking mass spectrometry (CX-MS) experiments show the stabilization of GTPγS-bound CtRoco in its elongated active conformation by Nb_Roco1 and/or Nb_Roco2. (**A**) Intramolecular disuccinimidyl sulfoxide (DSSO)-crosslinks within GTPγS-bound CtRoco in the absence of nanobodies. (**B**) Effect of Nb_Roco1 on the intramolecular DSSO-crosslinks within GTPγS-bound CtRoco. The intermolecular crosslinks between CtRoco and Nb_Roco1 are also shown. (**C**) Effect of Nb_Roco2 on the intramolecular DSSO-crosslinks within GTPγS-bound CtRoco. The intermolecular crosslinks between CtRoco and Nb_Roco2 are also shown. (**D**) Effect of the combination of Nb_Roco1 and Nb_Roco2 on the intramolecular DSSO-crosslinks within GTPγS-bound CtRoco. The intermolecular crosslinks between CtRoco, Nb_Roco1, and Nb_Roco2 are also shown.

The online version of this article includes the following figure supplement(s) for figure 4:

**Figure supplement 1.** Observed crosslinks between Nb_Roco1 and CtRoco.

previously could not detect any binding to the nucleotide-free CtRoco dimer (*Leemans et al., 2020*). The current cryo-EM structure shows that binding of Nb_Roco2 to nucleotide-free CtRoco must necessarily evoke a conformational change of the LRR. If this conformational change is relevant for the CtRoco activation mechanism and monomerization, we reasoned that binding of Nb_Roco2 might sufficiently weaken the CtRoco closed conformation to allow subsequent binding of Nb_Roco1. To test this hypothesis, we titrated FITC labeled Nb_Roco1 with increasing concentrations of nucleotide-free CtRoco that was pre-incubated with an excess of Nb_Roco2, and we followed the binding using fluorescence anisotropy measurements (*Figure 5A*). This shows that the presence of Nb_Roco2 enables Nb_Roco1 to bind to CtRoco in the absence of nucleotides, with a $K_d$ = 4.6 ± 1.6 µM. Hence, this clearly demonstrates a

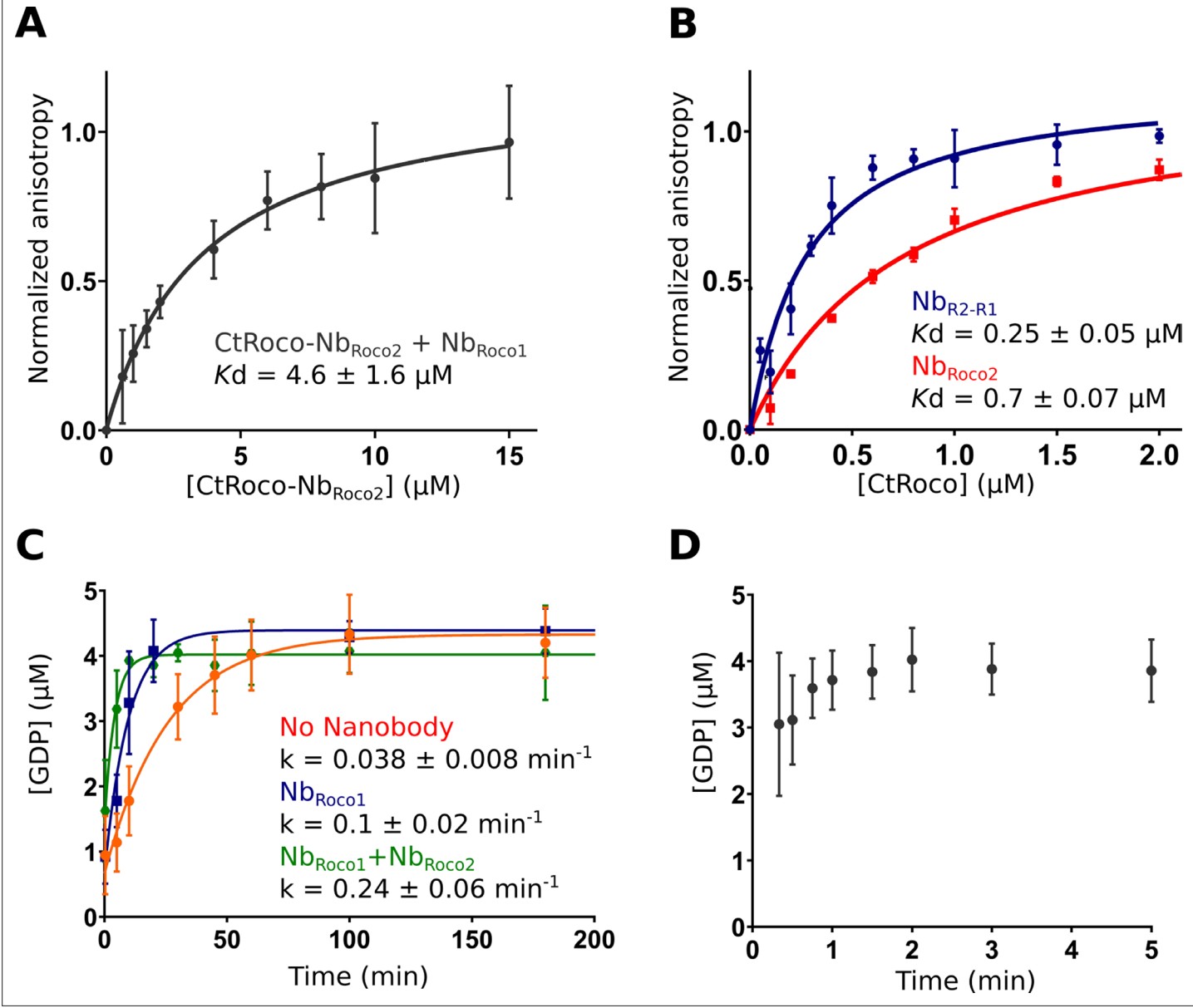

**Figure 5.** $Nb_{Roco1}$ and $Nb_{Roco2}$ have a synergistic effect on CtRoco activation that is potentiated by the bivalent nanobody construct $Nb_{R2-R1}$. (**A**) $Nb_{Roco2}$ enables the binding of $Nb_{Roco1}$ to the nucleotide-free CtRoco. The fluorescence anisotropy signal of the FITC-labeled $Nb_{Roco1}$ is monitored upon titration with increasing concentrations of nucleotide-free CtRoco in the presence of an excess of $Nb_{Roco2}$. The equilibrium dissociation constant ($K_d \pm$ standard error), obtained by fitting of triplicated data with a quadratic binding equation, is given. (**B**) $Nb_{R2-R1}$ has a threefold increased affinity toward nucleotide-free CtRoco in comparison to $Nb_{Roco2}$. FITC-labeled $Nb_{R2-R1}$ or $Nb_{Roco2}$ were titrated with nucleotide-free CtRoco, and fluorescence anisotropy data was analyzed as in (**A**). (**C**) Single turnover GTP (5 μM) hydrolysis rate of CtRoco-L487A (5 μM) in the absence of nanobodies (orange), or in the presence of either $Nb_{Roco1}$ (blue) or both $Nb_{Roco1}$ and $Nb_{Roco2}$ (green). (**D**) Single turnover GTP (5 μM) hydrolysis of CtRoco-L487A (5 μM) in the presence of $Nb_{R2-R1}$, all GTP was converted within the dead time (20 s) of the experiment.

synergistic effect of both Nbs in inducing conformational changes that drive CtRoco toward the active monomeric state.

## A bivalent $Nb_{Roco1}$-$Nb_{Roco2}$ construct ($Nb_{R2-R1}$) acts as a strong activator of CtRoco

Considering the mutual synergistic effect of $Nb_{Roco1}$ and $Nb_{Roco2}$, we reasoned that a bivalent/bi-paratopic nanobody construct that covalently links $Nb_{Roco1}$ to $Nb_{Roco2}$ would further increase the affinity and

avidity toward CtRoco. Based on our cryo-EM structure, we linked the C-terminal end of $Nb_{Roco2}$ to the N-terminus of $Nb_{Roco1}$ using a $(GGGGS)_3$ linker, hence obtaining the bi-paratopic construct $Nb_{R2-R1}$. As could be expected, $Nb_{R2-R1}$ shows binding to nucleotide-free CtRoco with a $K_d$ = 0.25 ± 0.05 µM (*Figure 5B*). This affinity of the bi-paratopic construct is threefold higher than that of $Nb_{Roco2}$ alone, and even 18-fold higher than that of $Nb_{Roco1}$ in the presence of an excess of $Nb_{Roco2}$. Thus, this clearly shows the cooperative effect of both Nbs in the bi-paratopic construct with regard to binding and inducing conformational changes in CtRoco.

Next, we wondered whether both Nbs would also cooperate in increasing the GTPase activity of CtRoco. Indeed, we previously showed that $Nb_{Roco1}$ increases the GTPase activity of the slow CtRoco-L487A mutant (*Leemans et al., 2020*). L487 is located in the Roc Switch 1 loop and the mutation is orthologous to the I1371V PD mutation in LRRK2 (*Jagtap et al., 2022*; *Paisán-Ruíz et al., 2005*). Therefore, we first performed single-turnover GTPase experiments by mixing 5 µM CtRoco-L487A with 5 µM GTP in the presence of an excess of both $Nb_{Roco1}$ and $Nb_{Roco2}$. In the presence of both Nbs, the hydrolysis rate was increased sixfold compared to CtRoco-L487A alone and twofold compared to CtRoco-L487A in the presence of $Nb_{Roco1}$ only, again illustrating a collaboration between the Nbs (*Figure 5C*). Finally, the single turnover experiment was performed in the presence of the bi-paratopic $Nb_{R2-R1}$. Interestingly, $Nb_{R2-R1}$ increases the GTPase turnover rate up to a level where it becomes too fast to determine an accurate $k_{obs}$ value. Indeed, while it takes the L487A mutant more than 1 hr to convert all the GTP under the conditions used, in the presence of $Nb_{R2-R1}$ nearly all the GTP has been hydrolyzed at the earliest possible sample point (20 s) (*Figure 5D*). This clearly illustrates the very strong cooperative activating effect of $Nb_{Roco1}$ and $Nb_{Roco2}$ within the bi-paratopic construct.

## Discussion

In this study, we present, to the best of our knowledge, the first structure of a protein belonging to the Roco family in a GTP-bound state, allowing us to analyze the conformational changes linked to nucleotide binding. While most attention has been devoted to human LRRK2 due to its link with PD, we used a bacterial homolog from the bacterium *C. tepidum* (CtRoco), which consists of the LRR, Roc, and COR (CORA/CORB) domains, followed by a C-terminal region of about 150 residues with unknown structure. Although this protein lacks the important kinase domain, it has proven to be an excellent model to study the properties of the generally conserved core LRR-Roc-COR domains that bear the GTPase activity, revealing the GTPase-driven dimer–monomer cycle and the kinetics of GTP hydrolysis (*Deyaert et al., 2017b*; *Wauters et al., 2018*). Previously, we solved an X-ray crystal structure of this protein in its nucleotide-free state, which displayed a compact dimeric arrangement with the LRR domain folding back on the Roc-COR domains within each protomer. Thus far, however, we have failed to produce well-diffracting crystals of this protein in a GTP-bound state, most probably due to the highly dynamic nature of the protein. To decrease this flexibility, we have now used two previously developed conformation-specific Nbs ($Nb_{Roco1}$ and $Nb_{Roco2}$) to stabilize the protein in the GTP-state (*Leemans et al., 2020*), allowing us to solve its structure using cryo-EM (*Figure 1*). Recently, Nbs have successfully been used to obtain structural insights into the conformational states of a number of highly dynamic proteins (*Uchański et al., 2020*). These studies established that Nbs bind antigens primarily by conformational selection rather than by induced fit (*Manglik et al., 2017*; *Smirnova et al., 2015*). Since $Nb_{Roco1}$ and $Nb_{Roco2}$ were generated by immunization with fully native CtRoco bound to a non-hydrolyzable GTP analog, and subsequently selected by phase display using the same functional protein, it is thus safe to assume that these Nbs bind to and stabilize a relevant conformation that is present within the 'active' CtRoco conformational space (*Leemans et al., 2020*). Moreover, our current structures are also in very good agreement with previous biochemical studies and data from HDX-MS and negative stain EM (*Deyaert et al., 2019*; *Deyaert et al., 2017b*).

The cryo-EM structure shows the GTPγS-bound CtRoco as an elongated monomer, where the LRR domain has undergone an approximately 135° rotation away from the Roc and COR domains. A comparison of the nucleotide-free CtRoco dimer with the GTPγS-bound monomer suggests a direct and reciprocal link between monomerization and the conformational changes within the protomers, and allows us to propose several concerted mechanisms linking both events (*Figure 2*, *Figure 2— figure supplement 2*). An initial trigger for monomerization seems to be the reorganization of the P-loop and Switch 2 region upon nucleotide binding, which is incompatible with the Roc:Roc dimer interface that is observed in the crystal structure of the nucleotide-free CtRoco. Via the Switch 2 loop

the conformational changes can be relayed, via the region linking the LRR with the Roc domain and the CORA, toward CORB (*Figure 2*). Specifically, in the GTP-state the CORB domain makes a 30° rotational movement in the direction of the CORB-CORB dimer interface of the CtRoco dimer. Such an orientation would result in severe steric clashes with the Roc domain of the adjacent protomer, which probably forms the major driving force for nucleotide-induced monomerization (*Figure 2*, *Figure 2—figure supplement 3*). Interestingly, the pivot point for the CORB rotational movement coincides with the hydroxyl group of Y804, which forms H-bonds with H554 and Y558 within helix $\alpha_3$ of the Roc domain (*Figure 2*). H554, Y558, and Y804 correspond to N1437, R1441, and Y1699 in LRRK2, respectively, and mutation of either of these three residues is associated with PD and LRRK2 kinase activation (*Kalogeropulou et al., 2022*). In LRRK2, a 'seesaw-like' movement of the $\alpha_3$-helix of Roc with respect to the CORB domain was observed upon activation by Rab29, and the N1437-R1441-Y1699 triad was proposed as a central region in the activation mechanism of LRRK2 (*Störmer et al., 2023*; *Zhu et al., 2023*). This thus illustrates that similar domain movements occur in LRRK2 and CtRoco, and that the GTPγS-induced conformational changes that we observe are thus likely also relevant to LRRK2.

The second prominent conformational change that occurs upon GTPγS binding is the movement of the LRR domain away from the Roc and COR domains. We hypothesize that also this conformational change is initially triggered by changes in the Switch 2 region upon GTP binding. In the nucleotide-free state, the backside of Switch 2 interacts with the LRR domain and the LRR-Roc linker via residues F528 and R532 (*Deyaert et al., 2019*). Rearrangement of Switch 2 upon nucleotide binding would disrupt these interactions. A key element for these conformational changes is the linker region between the LRR and Roc domains, with the $\alpha_0$-helix that immediately precedes the Roc domain playing a central role. While the $\alpha_0$-helix is buried in between the LRR, Roc, and CORA domain in the dimer, it loses all interactions with these domains in the GTPγS-bound monomer and acts as the swivel point for the rotational movement of the LRR domain (*Figure 2*). The conformational changes in the LRR domain upon nucleotide-induced activation of CtRoco are reminiscent of the observed changes in this domain upon activation of LRRK2 by Rab29 (*Zhu et al., 2023*). Indeed, in the kinase-inactive state of LRRK2, the LRR domain wraps around the kinase domain hence obstructing the entry of bulky substrates in the ATP pocket (*Myasnikov et al., 2021*). However, upon activation by Rab29 the N-terminal domains, including the LRR, become disordered liberating the access to the active site of the kinase domain. Superposition of the inactive CtRoco dimer with inactive LRRK2 shows a very similar position of their LRR domains and $\alpha_0$-helices, suggesting also comparable conformational changes upon activation

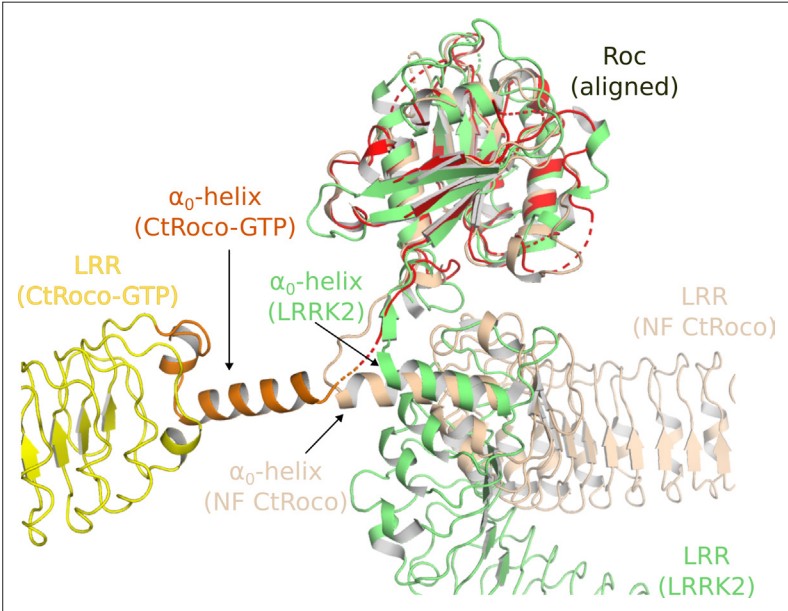

**Figure 6.** Similarities in the activation mechanisms of CtRoco and LRRK2. A comparison of the conformation of the LRR domain and the $\alpha_0$-helix adopted within GTPγS-bound CtRoco-Nb$_{Roco1}$-Nb$_{Roco2}$ (this work, colored by domain), nucleotide-free CtRoco (beige) (PDB: 6hlu) (*Deyaert et al., 2019*), and GDP-bound LRRK2 (green) (PDB: 7lhw) (*Myasnikov et al., 2021*) is shown. The Roc domain of the three structures was used for the superposition.

(*Figure 6*). In agreement with these findings, recent molecular dynamics simulations proposed a key role for the LRR-Roc linker in the activation of LRRK2 (*Weng et al., 2023*). Moreover, a R1325Q variant in the $\alpha_0$-helix of LRRK2 was recently identified as a kinase-activating PD mutation (*Kalogeropulou et al., 2022*). In LRRK2, R1325 interacts with F1284, N1286, and N1305 located at the C-terminus of the LRR domain and with P1524 located in the linker between Roc and CORA, and one could imagine that the R1325Q mutation would weaken the interaction between the LRR and Roc-COR domains and activate the kinase. The above mechanisms thus suggest a strong coupling between CtRoco monomerization and the conformational changes within each protomer. Such a link between monomerization and activation was recently also suggested in LRRK2 by the finding that mutations in the CORB:CORB dimer interface (R1731L/R1731D) enhance LRRK2 kinase activity (*Snead et al., 2022*).

Next to these insights into the nucleotide-mediated activation mechanism of Roco proteins, our structure also reveals how $Nb_{Roco1}$ and $Nb_{Roco2}$ can allosterically activate the GTPase activity of CtRoco. The most prominent role in this regard seems to be for $Nb_{Roco1}$, which binds in the region of CtRoco that undergoes the largest conformational changes upon GTP-driven monomerization, consisting of the linker regions connecting the LRR to the Roc domain and the Roc to the CORA domain (*Figure 3*). The extended conformation of the CtRoco monomer is further stabilized by the binding of $Nb_{Roco2}$ in the curvature of the horseshoe-shaped LRR domain, which would sterically exclude the LRR conformation adopted in the compact CtRoco dimer. We also find that $Nb_{Roco1}$ and $Nb_{Roco2}$ cooperate in shifting the CtRoco dimer–monomer equilibrium toward the activated monomer, since binding of $Nb_{Roco2}$ to nucleotide-free CtRoco allows the subsequent binding of $Nb_{Roco1}$. This again proves that the conformational changes occurring upon CtRoco activation are closely linked and act in a concerted manner. Correspondingly, the bivalent/bi-paratopic $Nb_{R2-R1}$ construct shows an increased affinity towards CtRoco and a very strong stimulation of the GTPase activity of the slow CtRoco-L487A mutant, illustrating the power of the cooperative avidity effects enabled by such bivalent constructs. Considering that the conformational changes of the LRR domain, and the role of the LRR-Roc linker in this regard, are conserved features within the activation mechanisms of CtRoco and LRRK2, one could imagine that a similar mode of action could also be applicable to Nbs binding to LRRK2. Previously, we described Nbs that can allosterically inhibit or activate the kinase activity of human LRRK2, but the exact mechanism underlying this activity is still elusive, and it is unclear whether these Nbs use similar mechanisms to the GTPase-activating Nbs described here (*Singh et al., 2022*). Such a set of Nbs that either modify the GTPase or kinase activity of LRRK2 would form exquisite tools to aid in unraveling the link between both catalytic activities, as well as the role of the GTPase in LRRK2 pathology.

In conclusion, this study provides important new structural insights into the mechanism of activation of Roco proteins with relevance for LRRK2, suggests new avenues for allosteric modulation of Roco activity, and underscores the remaining open questions with regard to the communication between GTPase and kinase activities in LRRK2.

## Materials and methods
### Cloning of the bivalent nanobody construct
To generate the bivalent $Nb_{R2-R1}$, a gene construct was designed that fuses the $Nb_{Roco2}$ to the $Nb_{Roco1}$ open-reading frame with the following linker sequence:

5'-GGCGGCGGCGGCAGCGGCGGCGGCGGCAGCGGCGGCGGCGGCAGC-3'.

The gene was synthesized and subcloned by GenScript Biotech (The Netherlands) in the pHEN29 vector, which adds the Sortase-recognition sequence LPETG at the C-terminus, allowing for Sortase A-mediated labeling. The plasmid was initially transformed in *Escherichia coli* DH5α cells and subsequently in WK6 cells for protein expression.

### Protein expression and purification
CtRoco, CtRoco-L487A, $Nb_{Roco1}$, and $Nb_{Roco2}$ were expressed and purified as previously described (*Leemans et al., 2020*). In brief, CtRoco and CtRoco-L487A were produced with an N-terminal His-tag from the pProEX plasmid in an *E. coli* BL21(DE3) strain (genotype: F⁻ *hsdS*ᴮ (rᴮ⁻ mᴮ⁻) *gal dcm* (DE3) pLysS (Cmᴿ)). For purification, the protein was first subjected to a $Ni^{2+}$-NTA immobilized metal affinity chromatography step, using a His-trap FF column. After elution, the protein was dialyzed against 20 mM HEPES/NaOH pH 7.5, 150 mM NaCl, 5% glycerol, 1 mM DTT, and 1 mM EDTA, which were

added to the protein to disrupt $Mg^{2+}$ and nucleotide binding. As a second purification step, an SEC on a Superdex 200 column (GE Healthcare) was performed using the same buffer. After gel filtration, 5 mM $MgCl_2$ was added to the protein sample and analytical reversed-phase chromatography was used to confirm the complete removal of nucleotides, as described previously (**Deyaert et al., 2017b**). Samples were concentrated and flash frozen until use.

$Nb_{Roco1}$ and $Nb_{Roco2}$ and $Nb_{R2-R1}$ were produced in *E. coli* WK6 cells (genotype: Δ(*lac-proAB*) *galE strA/F'* lacI$^q$ lacZΔM15 proA$^+$B$^+$) with either a C-terminal His-tag from a pMESy4 vector (for structural studies), or with a C-terminal Sortase-His-tag from a pHEN29 vector (for fluorescent labeling). After an osmotic shock to obtain the periplasmic fraction, an affinity purification step on $Ni^{2+}$-NTA sepharose was performed followed by SEC on a Superdex 75 column equilibrated with the same buffer as used for the CtRoco protein.

## Sample preparation and cryo-EM data acquisition

CtRoco was loaded with GTPγS by incubation with 0.5 mM of the nucleotide. Subsequently, the GTPγS-loaded protein was incubated with either $Nb_{Roco1}$ alone (CtRoco-$Nb_{Roco1}$) or with $Nb_{Roco1}$ and $Nb_{Roco2}$ (CtRoco-$Nb_{Roco1}$.$Nb_{Roco2}$) using a 2× molar excess of the Nbs. An SEC was performed using a Superdex200 column to separate the Nb excess from the complex, the sample was immediately supplemented again with an excess of GTPγS, and the concentration of the sample was adjusted to 0.08 mg/ml. Quantifoil (2/1) 300-mesh copper Holey grids were glow-discharged for 30 s to 1 min, and 3 μl of the complex was loaded on the grid and blotted for 1 s using Whatman paper, before being frozen in liquid ethane on a Cryoplunge3. Grids were stored in liquid nitrogen until use.

Single-particle cryo-EM data were collected on a JEOL CryoARM300 transmission-electron microscope, operated at 300 kV, and at a nominal magnification of 60,000 and corresponding pixel size of 0.76 Å. The microscope contained an omega energy filter with a slit width set to 20 eV. The images were recorded using a K3 detector (Gatan) operating in correlative-double sampling (CDS) mode. Micrographs were recorded as movies of 60 frames using SerialEM v3.0.8. A total of 11,718 movies were collected for the CtRoco-$Nb_{Roco1}$ data set and 7489 movies were collected for the CtRoco-$Nb_{Roco1}$-$Nb_{Roco2}$ data set. Data collection statistics are reported in **Table 1**.

## Image processing

Data were processed on the fly using Relion 3.1 (**Zivanov et al., 2018**) including gain normalization, motion correction, and calculation of dose-weighted averages with UCSF MotionCor2 (**Zheng et al., 2017**). Results were analyzed and curated in the first instance using the in-house script BXEMALYZER (Shkumatov et al., in preparation). The motion-corrected micrographs were imported into cryoSPARC v4.3 (**Punjani et al., 2017**) and CTF was calculated using Patch CTF.

For the CtRoco-$Nb_{Roco1}$ data set, particle picking was first performed using manual picking followed by template picker (using initial 2D classes). After several rounds of 2D classification, an ab initio reconstruction was obtained. Particle stacks from 2D classes that showed clear features of the full complex were selected for further processing, and together with the initial map imported into Relion 3.1. 3D auto-refine was performed in Relion, aligned particle stacks and models were imported back to cryoSPARC, and particles re-extracted with updated particle coordinates; this procedure improved the centering of the particles, resulting in more classes displaying the full particle. A heterogeneous refinement with four classes, followed by homogeneous refinement of the best class based on density interpretability, yielded an 8.3 Å resolution map reconstruction from 99,460 particles (**Figure 1—figure supplement 2**).

For the CtRoco-$Nb_{Roco1}$-$Nb_{Roco2}$ data set, particle picking was first performed with Topaz (**Bepler et al., 2019**), after import of the micrographs to cryoSPARC, using the pretrained model ResNet16 on a small set of 18 random images. Bad particles were removed by 2D classification and the rest were used to train a Topaz model, followed by picking using Topaz Extract on the full data set, which resulted in 716,919 particles extracted on a box size of 389 Å. After three rounds of 2D classification, 371,956 particles were selected. Selected particles were used for multiple ab initio reconstruction using six classes, followed by heterogeneous refinement of the six classes. These jobs yielded one class corresponding to the volume of CtRoco bound to both Nbs and two classes corresponding to good-quality volumes centered on the LRR domain but with little density for the other half of the molecule. The other three classes were also centered on the LRR domain but with overall bad

quality. Homogeneous refinement of the class displaying a nearly full CtRoco protein bound to the two Nbs, followed by non-uniform refinement (*Punjani et al., 2020*), resulted in a 7.7 Å resolution map reconstruction from 48,333 particles. To obtain a focused map on the LRR half of the protein, the two classes centered on the LRR domain were merged, particles were re-extracted on a smaller box size of 243 Å, and, after homogeneous and non-uniform refinements, a map was obtained where only the LRR domain bound to Nb$_{Roco2}$ is visible with 3.6 Å resolution using 160,925 particles. To obtain a higher resolution map for the other half of the protein, spanning the Roc-COR and Nb$_{Roco1}$, a different approach was used: three 2D classes centered on the Roc-COR domains were selected and fed to a template picker job with the intention to pick this half of the particles only, and extracted on a box size of 243 Å. After extraction and three rounds of 2D classification, again only classes that were centered on the Roc-COR domains were selected for ab initio reconstruction using four classes followed by heterogeneous refinement. These yielded one good-quality volume and the particles in this class were then subjected to non-uniform refinement, yielding a map representing the Roc-COR domains and the Nb$_{Roco1}$. The overall resolution of the map was estimated at 3.9 Å using 38,260 particles (*Figure 1—figure supplement 1*).

## Model building and refinement

The focused maps containing either the LRR domain bound to Nb$_{Roco2}$ or the Roc-COR domain bound to Nb$_{Roco1}$ were improved by density modification with resolve_cryo_em (*Terwilliger et al., 2020*) in the PHENIX suit (version 1.20.1) (*Adams et al., 2010*), and used for model interpretation and refinement. Initial models of the individual CtRoco domains were taken from the nucleotide-free CtRoco structure (PDB: 6hlu). Nb models were generated using AlphaFold2 (*Jumper et al., 2021*). Initial rigid body fits were performed for each domain separately on UCSF ChimeraX 1.2 (*Pettersen et al., 2021*), followed by a flexible fitting in Coot 0.9.2 (*Emsley et al., 2010*), to fit the secondary structure features into density. Manual building and inspection was also done in Coot, and real-space refinement was performed with the phenix.real_space_refine program in PHENIX applying Ramachandran plot restraints. Model-versus-data fit was assessed by curves of the Fourier shell correlation (FSC) as a function of resolution using the refined models and the two half-maps of each focused volume (*Figure 1—figure supplement 5*). Finally, the focused maps and models were fitted into the lower resolution map of the entire protein in UCSF ChimeraX to obtain an overall interpretation of CtRoco-Nb$_{Roco1}$-Nb$_{Roco2}$. Figures containing molecular structures and volumes were prepared with UCSF ChimeraX and PyMOL (The PyMol Molecular Graphic System, version 2.4 Schrödinger, LLC, https://pymol.org/2/). Maps shown in the figures were improved with either Resolve or EMReady (*He et al., 2023*).

## Chemical crosslinking mass spectrometry

CX-MS was performed as previously described (*Singh et al., 2022*). Briefly, the CtRoco concentration was adjusted to 8 µM in storage buffer. GTPγS-bound CtRoco was crosslinked with DSSO at a molar ratio of 1:50 (protein: crosslinker). In addition, GTPγS-bound CtRoco was mixed with either Nb$_{Roco1}$, Nb$_{Roco2}$ or both in a molar ratio 1:5 (CtRoco:Nb) before DSSO crosslinking (molar ratio 1:50). The reaction was carried on for 30 min at room temperature and stopped by adding 10 µl of 1 M Tris at pH 7.5. Proteins were precipitated by chloroform/methanol and subjected to tryptic proteolysis. The tryptic peptide solutions were cleaned up by C18-StageTips (Thermo Fisher) and the volume was reduced to approximately 10 µl in a SpeedVac. 40 µl SEC buffer (30% [vol/vol] acetonitrile, 0.1% TFA) was added to the desalted peptides. The entire volume of 50 µl was loaded onto the SEC column (Superdex Peptide column 3.2/300; Cytiva), which was mounted to an Äkta pure system (Cytiva) and equilibrated in SEC buffer. SEC was performed at a flow rate of 50 µl/min. The eluates were collected in 100 µl fractions. Vacuum-dried fractions (remaining volume of 2 µl to avoid complete dryness) containing the crosslinked peptides, were redissolved in a total volume of 10 µl 0.5% TFA, and analyzed individually on an Orbitrap Fusion mass spectrometer (Thermo Fisher) using the MS2_MS3 fragmentation method with the default settings (version 3.4, build 3072). MS1 scans were performed in the Orbitrap (FTMS, resolution = 60K) at an m/z range of 375–1500. MS2 was performed with CID (CE = 25%) and spectra were acquired in the Orbitrap (FTMS) at 30K resolution. The MS3 scans were performed with HCD (CE = 30%) and spectra were acquired in the linear ion trap. Resulting Thermo Raw files were analyzed

with the MS2_MS3 workflow provided by in Proteome Discoverer 2.5 (build 2.5.0.400) using XlinkX (version 2.5).

## Fluorescence anisotropy titrations

The affinity of $Nb_{Roco1}$, $Nb_{Roco2}$, and $Nb_{R2-R1}$ for CtRoco was determined using fluorescence anisotropy titrations, Nbs were labeled at their C-terminus with a FITC fluorophore using Sortase A-mediated labeling, as previously described (*Leemans et al., 2020*). Fluorescence anisotropy titrations with FITC-labeled Nbs and CtRoco were performed at 25°C using a Cary Eclipse spectrofluorometer (Agilent) equipped with polarizers and temperature control, at an excitation wavelength of 493 nm and emission wavelength of 516 nm. 50 nM FITC-labeled Nb was titrated with increasing amounts of CtRoco. The anisotropy signal was measured after a 2 min incubation period for each titration. All experiments were performed in triplicate. To obtain $K_d$ values (± standard error), the data were fitted using the quadratic binding equation in GraphPad Prism 7.

## Single-turnover kinetics

GTP hydrolysis rates were calculated under single-turnover conditions. 5 µM of CtRoco-L487A was incubated with 5 µM of GTP in the presence of 100 µM of the Nb of interest at 25°C. Samples were taken at different time points ranging from 0 to 180 min and the reaction was stopped by incubation at 95°C for 3 min. Samples were mixed with the same volume of HPLC buffer and 50 µl was injected on a reversed-phase C18 column (Phenomenex, Jupiter 5 mm) attached to an Alliance e2695 HPLC (Waters) using 100 mM $KH_2PO_4$ pH 6.4, 10 mM tetrabutylammonium bromide, 7.5% acetonitrile as the mobile phase. The 254 nm absorption peaks of GDP were converted to concentrations by using a standard curve, and GDP concentrations were plotted in function of time. All experiments were performed in triplicate and data were fitted on single-exponential equation using GraphPad Prism 7.

## Acknowledgements

We thank Siemen Claeys for excellent technical support, Jonathan Mitano for his assistance in some of the experiments, and all members of the Versées lab for comments and discussions. We also thank Dirk Reiter from the BECM cryo-EM facility for EM computational support. The authors wish to thank the staff of the Core Facility for Medical Proteomics (Tübingen) for technical assistance. This work was supported by grants from the Research Foundation Flanders (G005219N, G003322N) and a Strategic Research Program Financing from the VUB (SRP50, SRP95).

## Additional information

### Funding

| Funder | Grant reference number | Author |
| --- | --- | --- |
| Fonds Wetenschappelijk Onderzoek | G005219N | Wim Versées |
| Fonds Wetenschappelijk Onderzoek | G003322N | Wim Versées |
| Vrije Universiteit Brussel | SRP50 | Wim Versées |
| Vrije Universiteit Brussel | SRP95 | Wim Versées |

The funders had no role in study design, data collection and interpretation, or the decision to submit the work for publication.

### Author contributions

Christian Galicia, Data curation, Formal analysis, Investigation, Methodology, Writing – original draft, Writing – review and editing; Giambattista Guaitoli, Data curation, Investigation; Marcus Fislage, Supervision, Methodology, Writing – review and editing; Christian Johannes Gloeckner, Formal analysis, Investigation, Writing – original draft, Writing – review and editing; Wim Versées, Conceptualization,

Resources, Formal analysis, Supervision, Funding acquisition, Validation, Investigation, Methodology, Writing – original draft, Project administration, Writing – review and editing

### Author ORCIDs
Christian Galicia http://orcid.org/0000-0001-6080-7533
Marcus Fislage http://orcid.org/0000-0002-2527-2657
Christian Johannes Gloeckner http://orcid.org/0000-0001-6494-6944
Wim Versées http://orcid.org/0000-0002-4695-696X

Reviewer #1 (Public Review): https://doi.org/10.7554/eLife.94503.3.sa1
Reviewer #2 (Public Review): https://doi.org/10.7554/eLife.94503.3.sa2
Author response https://doi.org/10.7554/eLife.94503.3.sa3

# Additional files

### Supplementary files
• MDAR checklist

### Data availability
Cryo-EM Density maps and structure coordinates have been deposited in the Electron Microscopy Data Bank (EMDB) and the Protein Data Bank (PDB), with accession codes EMD-18884 and PDB 8R4B for the composite map and structure of CtRoco-$Nb_{Roco1}$-$Nb_{Roco2}$, EMD-18885 and PDB 8R4C for the LRR-$Nb_{Roco2}$ focused map and structure, EMD-18886 and PDB 8R4D for the Roc-COR-$Nb_{Roco1}$ focused map and structure, EMD-18882 for the low resolution full CtRoco-$Nb_{Roco1}$-$Nb_{Roco2}$ map, and EMD-18879 for the CtRoco-$Nb_{Roco1}$ map. The XL-MS data have been deposited to the ProteomeXchange Consortium via the PRIDE (*Perez-Riverol et al., 2019*) partner repository with the dataset identifier PXD046634 and DOI:10.6019/PXD046634. Source data for binding and kinetics experiments were deposited on the online repository Zenodo with identifier 10.5281/zenodo.10718509. Materials can be obtained from the Versées Lab by contacting mta.requests@vib.be or wim.versees@vub.be.

The following datasets were generated:

| Author(s) | Year | Dataset title | Dataset URL | Database and Identifier |
| --- | --- | --- | --- | --- |
| Galicia C, Versees W | 2024 | Ensemble map of the Roco protein from C. tepidum in the GTP state bound to the activating Nanobodies NbRoco1 and NbRoco2 | https://www.ebi.ac.uk/emdb/EMD-18884 | Electron Microscopy Data Bank, EMD-18884 |
| Galicia C, Versees W | 2024 | Focused map on the LRR domain of the Roco protein from *C. tepidum* bound to the activating Nanobody NbRoco2 | https://www.ebi.ac.uk/emdb/EMD-18885 | Electron Microscopy Data Bank, EMD-18885 |
| Galicia C, Fislage M, Versées W | 2024 | Roco protein from C. tepidum in the GTP state bound to an activating Nanobody | https://www.ebi.ac.uk/emdb/EMD-18879 | Electron Microscopy Data Bank, EMD-18879 |
| Galicia C, Fislage M, Versées W | 2024 | Roco protein from C. tepidum in the GTP state bound to the activating Nanobodies NbRoco1 and NbRoco2 | https://www.ebi.ac.uk/emdb/EMD-18882 | Electron Microscopy Data Bank, EMD-18882 |

*Continued on next page*

*Continued*

| Author(s) | Year | Dataset title | Dataset URL | Database and Identifier |
|---|---|---|---|---|
| Galicia C, Versées W | 2024 | Structural insights in the GTP-driven monomerization and activation of a bacterial LRRK2 homologue using allosteric nanobodies | https://doi.org/10.5281/zenodo.10718509 | Zenodo, 10.5281/zenodo.10718509 |
| Galicia C, Versées W | 2024 | Roco protein from C. tepidum in the GTP state bound to the activating Nanobodies NbRoco1 and NbRoco2 | https://www.rcsb.org/structure/8R4B | RCSB Protein Data Bank, 8R4B |
| Galicia C, Versées W | 2024 | LRR domain of Roco protein from C. tepidum bound to the activating Nanobody NbRoco2 | https://www.rcsb.org/structure/8R4C | RCSB Protein Data Bank, 8R4C |
| Galicia C, Versées W | 2024 | Focused map on the Roc-COR domains of the Roco protein from C. tepidum in the GTP state bound to the activating Nanobody NbRoco1 | https://www.ebi.ac.uk/emdb/EMD-18886 | EMDB, EMD-18886 |
| Galicia C, Versées W | 2024 | Focused map on the Roc-COR domains of the Roco protein from C. tepidum in the GTP state bound to the activating Nanobody NbRoco1 | https://www.rcsb.org/structure/8R4D | RCSB Protein Data Bank, 8R4D |
| Gloeckner CJ | 2024 | Analysis of conformational changes induced by binding of allosteric nanobodies to a bacterial LRRK2 homologue by CX-MS | https://www.ebi.ac.uk/pride/archive/projects/PXD046634 | PRIDE, PXD046634 |

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
