## [Editor Report · eLife assessment]

The **fundamental** study by Galicia et al. captured the GTP-bound active structure of CtRoco, a homolog of human LRRK2, using conformation-specific nanobodies. This **convincing** body of work reports the first structure of a GTP-bound ROCO protein, illustrating how GTP facilitates the dimer-to-monomer transition of CtRoco and functional activation.

---

## [Referee Report · Reviewer #1 (Public Review)]

Summary:

The Roco proteins are a family of GTPases characterized by the conserved presence of an ROC-COR tandem domain. How GTP binding alters the structure and activity of Roco proteins remains unclear. In this study, Galicia C et al. took advantage of conformation-specific nanobodies to trap CtRoco, a bacterial Roco, in an active monomeric state and determined its high-resolution structure by cryo-EM. This study, in combination with the previous inactive dimeric CtRoco, revealed the molecular basis of CtRoco activation through GTP-binding and dimer-to-monomer transition.

Strengths:

The reviewer is impressed by the authors' deep understanding of the CtRoco protein. Capturing Roco proteins in a GTP-bound state is a major breakthrough in the mechanistic understanding of the activation mechanism of Roco proteins and shows similarity with the activation mechanism of LRRK2, a key molecule in Parkinson's disease. Furthermore, the methodology the authors used in this manuscript - using conformation-specific nanobodies to trap the active conformation, which is otherwise flexible and resistant to single-particle average - is highly valuable and inspiring.

---

## [Referee Report · Reviewer #2 (Public Review)]

Summary

The manuscript by Galicia et al describes the structure of the bacterial GTPyS-bound CtRoco protein in the presence of nanobodies. The major relevance of this study is in the fact that the CtRoco protein is a homolog of the human LRRK2 protein with mutations that are associated with Parkinson's disease. The structure and activation mechanisms of these proteins are very complex and not well understood. Especially lacking is a structure of the protein in the GTP-bound state. Previously the authors have shown that two conformational nanobodies can be used to bring/stabilize the protein in a monomer-GTPyS-bound state. In this manuscript, the authors use these nanobodies to obtain the GTPyS-bound structure and importantly discuss their results in the context of the mammalian LRRK2 activation mechanism and mutations leading to Parkinson's disease. The work is well performed and clearly described. In general, the conclusions on the structure are reasonable and well-discussed in the context of the LRRK2 activation mechanism.

Strengths:

The strong points are the innovative use of nanobodies to stabilize the otherwise flexible protein and the new GTPyS-bound structure that helps enormously in understanding the activation cycle of these proteins.

---

## [Author Response]

The following is the authors’ response to the original reviews.

**Public Review:**

**Reviewer #1:**
Summary:The Roco proteins are a family of GTPases characterized by the conserved presence of an ROC-COR tandem domain. How GTP binding alters the structure and activity of Roco proteins remains unclear. In this study, Galicia C et al. took advantage of conformationspecific nanobodies to trap CtRoco, a bacterial Roco, in an active monomeric state and determined its high-resolution structure by cryo-EM. This study, in combination with the previous inactive dimeric CtRoco, revealed the molecular basis of CtRoco activation through GTP-binding and dimer-to-monomer transition.Strengths:The reviewer is impressed by the authors' deep understanding of the CtRoco protein. Capturing Roco proteins in a GTP-bound state is a major breakthrough in the mechanistic understanding of the activation mechanism of Roco proteins and shows similarity with the activation mechanism of LRRK2, a key molecule in Parkinson's disease. Furthermore, the methodology the authors used in this manuscript - using conformation-specific nanobodies to trap the active conformation, which is otherwise flexible and resistant to single-particle average - is highly valuable and inspiring.Weakness:Though written with good clarity, the paper will benefit from some clarifications.(1) The angular distribution of particles for the 3D reconstructions should be provided (Figure 1 - Sup. 1 & Sup. 2).

Figure 1 – Figure supplements 1 and 2 now contain particle distribution plots.

(2) The B-factors for protein and ligand of the model, Map sharpening factor, and molprobity score should be provided (Table 1).

Table 1 now contains B-factors and molprobity scores.

The map used to interpret the model was post-processed by density modification, and therefore no data concerning sharpening factors are provided in the output.

(3) A supplemental Figure to Figure 2B, illustrating how a0-helix interacts with COR-A&LRR before and after GTP binding in atomic details, will be helpful for the readers to understand the critical role of a0-helix during CtRoco activation.

This is now illustrated in the new Figure 2 – Figure Supplement 1.

(4) For the following statement, "On the other hand, only relatively small changes are observed in the orientation of the Roc a3 helix. This helix, which was previously suggested to be an important element in the activation of LRRK2 (Kalogeropulou et al., 2022), is located at the interface of the Roc and CORB domains and harbors the residues H554 and Y558, orthologous to the LRRK2 PD mutation sites N1337 and R1441, respectively." It is not surprising the a3-helix of the ROC domain only has small changes when the ROC domain is aligned (Figure 2E). However, in the study by Zhu et al (DOI: 10.1126/science.adi9926), it was shown that a3-helix has a "see-saw" motion when the COR-B domain is aligned. Is this motion conserved in CtRoco from inactive to active state?

We indeed describe the conformational changes from the perspective of the Roc domain. When using the COR-B domain for structural alignment, a rotational movement of Roc (including a “seesaw”-like movement of the α3-helix helix around His554) with respect to COR-B is correspondingly observed.

This is now added to Figure 2E. Additionally, the text was adapted to:

“Interestingly, this rotational movement of CORB seems to use the H554-Y558-Y804 triad on the interface of Roc and CORB as a pivot point (Figure 2E). Mutation of either of the corresponding residues in LRRK2 (N1437, R1441, Y1699, respectively) is associated with PD and leads to LRRK2 activation. Residues H554 and Y558 are located on the Roc a3 helix, which was previously suggested to be an important element in the activation of LRRK2 (Kalogeropulou et al., 2022). Indeed, while the orientation of the a3 helix with respect to the rest of the Roc domain only undergoes small changes upon GTPgS binding, it can be observed that this helix undergoes a “seesaw-like” movement with respect to the CORB domain. A similar rearrangement was previously also observed for Rab29-mediated activation of human LRRK2 (Störmer et al., 2023; Zhu et al., 2022).”

(5) A supplemental figure showing the positions of and distances between NbRoco1 K91 and Roc K443, K583, and K611 would help the following statement. "Also multiple crosslinks between the Nbs and CtRoco, as well as between both nanobodies were found. ... NbRoco1-K69 also forms crosslinks with two lysines within the Roc domain (K583 and K611), and NbRoco1-K91 is crosslinked to K583".

A figure displaying these crosslinks is now provided as Figure 4–figure supplement 1. However, in interpreting these crosslinks it should be taken into consideration that the additive length of the DSSO spacer and the lysine side chains leads to a theoretical upper limit of ∼26 Å for the distance between the α carbon atoms of cross-linked lysines (and even a cut-off distance of 35 Å when taking into account protein dynamics).

(6) It would be informative to show the position of CtRoco-L487 in the NF and GTP-bound state and comment on why this mutation favors GTP hydrolysis.

L487 is located in Switch 1, which is a critical region for nucleotide binding and hydrolysis. Unfortunately, most probably due to flexibility, the Switch 1 region could not be entirely modeled (in neither nucleotide state). Since L487 is located on the edge of the interpretable portion of the Switch 1 in both structures (see Author response image 1 below), any interpretation regarding the role of this residue would be highly speculative.

The following text was added to the Results section:

“Also the Switch 1 loop could not be fully modeled in our structure, presumably indicating some flexibility in this region despite the presence of a GTP analogue. Interestingly, the Switch 1 loop harbors the site of the PD-analogous L487A mutation that leads to a stabilization of the CtRoco dimer with a concomitant decrease in GTPase activity (Deyaert et al., 2019). Unfortunately, an exact interpretation of this effect of the L487A mutation is hampered by the lack of a well resolved Switch 1 loop.”

**Reviewer #2:**
SummaryThe manuscript by Galicia et al describes the structure of the bacterial GTPyS-bound CtRoco protein in the presence of nanobodies. The major relevance of this study is in the fact that the CtRoco protein is a homolog of the human LRRK2 protein with mutations that are associated with Parkinson's disease. The structure and activation mechanisms of these proteins are very complex and not well understood. Especially lacking is a structure of the protein in the GTP-bound state. Previously the authors have shown that two conformational nanobodies can be used to bring/stabilize the protein in a monomerGTPyS-bound state. In this manuscript, the authors use these nanobodies to obtain the GTPyS-bound structure and importantly discuss their results in the context of the mammalian LRRK2 activation mechanism and mutations leading to Parkinson's disease. The work is well performed and clearly described. In general, the conclusions on the structure are reasonable and well-discussed in the context of the LRRK2 activation mechanism.Strengths:The strong points are the innovative use of nanobodies to stabilize the otherwise flexible protein and the new GTPyS-bound structure that helps enormously in understanding the activation cycle of these proteins.Weakness:The strong point of the use of nanobodies is also a potential weak point; these nanobodies may have induced some conformational changes in a part of the protein that will not be present in a GTPyS-bound protein in the absence of nanobodies.Two major points need further attention.(1) Several parts of the protein are very flexible during the monomer-dimer activity cycle. This flexibility is crucial for protein function, but obviously hampers structure resolution. Forced experiments to reduce flexibility may allow better structure resolution, but at the same time may impede the activation cycle. Therefore, careful experiments and interpretation are very critical for this type of work. This especially relates to the influence of the nanobodies on the structure that may not occur during the "normal" monomerdimer activation cycle in the absence of the nanobodies (see also point 2). So what is the evidence that the nanobody-bound GTPyS-bound state is biochemically a reliable representative of the "normal" GTP-bound state in the absence of nanobodies, and therefore the obtained structure can be confidentially used to interpret the activation mechanism as done in the manuscript.

See below for an answer to remark 1 and 2.

(2) The obtained structure with two nanobodies reveals that the nanobodies NbRoco1 and NbRoco2 bind to parts of the protein by which a dimer is impossible, respectively to a0helix of the linker between Roc-COR and LRR, and to the cavity of the LRR that in the dimer binds to the dimerizing domain CORB. It is likely the open monomer GTP-bound structure is recognized by the nanobodies in the camelid, suggesting that overall the open monomer structure is a true GTP-bound state. However, it is also likely that the binding energy of the nanobody is used to stabilize the monomer structure. It is not automatically obvious that in the details the obtained nonobody-Roco-GTPyS structure will be identical to the "normal" Roco-GTPyS structure. What is the influence of nanobody-binding on the conformation of the domains where they bind; the binding energy may be used to stabilize a conformation that is not present in the absence of the nanobody. For instance, NbRoco1 binds to the a0 helix of the linker; what is here the "normal" active state of the Roco protein, and is e.g. the angle between RocCOR and LRR also rotated by 135 degrees? Furthermore, nanobody NbRoco2 in the LRR domain is expected to stabilize the LRR domain; it may allow a position of the LRR domain relative to the rest of the protein that is not present without nanobody in the LRR domain. I am convinced that the observed open structure is a correct representation of the active state, but many important details have to be supported by e,g, their CX-MS experiments, and in the end probably need confirmation by more structures of other active Roco proteins or confirmation by a more dynamic sampling of the active states by e.g. molecular dynamics or NMR.

Recently, nanobodies have increasingly been used successfully to obtain structural insights in protein conformational states (reviewed in Uchański et al, Curr. Opin. Struc. Biol. 2020). As reviewer # 2 points out, the concern is sometimes raised that antibodies could distort a protein into non-native conformations. Here, it is important to note that the nanobodies were raised by immunizing a llama with the fully native CtRoco protein bound to a non-hydrolysable GTP analogue, after which the nanobodies were selected by phage display using the same fully native and functional form of the protein. As clearly explained in Manglik et al. Annu Rev Pharmacol Toxicol. 2017, the probability of an in vivo matured nanobody inducing a non-native conformation of the antigen is low, although it is possible that it selects a high-energy, low-population conformation of a dynamic protein. Immature B cells require engagement of displayed antibodies with antigen to proliferate and differentiate during clonal selection. Antibodies that induce non-native conformations of the antigen pay a substantial energetic penalty in this process, and B cell clones displaying such antibodies will have a significantly lower probability of proliferation and differentiation into mature antibody-secreting B lymphocytes. Hence, many recent experiments and observation give credence to the notion that nanobodies bind antigens primarily by conformational selection and not induced fit (e.g. Smirnova et al. PNAS 2015).

Extrapolated to the case of CtRoco, which is clearly very flexible in its GTP-bound form, this means that the nanobodies are able to trap and stabilize one conformational state that is representative of the “active state” ensemble of the protein. In this respect, it is clear from our experiments (XL-MS, affinity and effect on GTPase activity) that the effects of NbRoco1 and NbRoco2 are additive (or even cooperative), meaning that both nanobodies recognize different features of the same CtRoco “active state”. Correspondingly, the monomeric, elongated “open” conformation is also observed in the structure of CtRoco bound to NbRoco1 only (Figure1 - supplement 2), albeit that this structure still displays more flexibility. The monomerization and conformational changes that we observe and describe in the current paper at high resolution are also in very good agreement with earlier observations for CtRoco in the GTP-bound form in absence of any nanobodies, including negative stain EM (Deyaert et al. Nature Commun, 2017), hydrogen-deuterium exchange experiments (Deyaert et al. Biochem. J. 2019) and native MS (Leemans et al. Biochem J. 2020).

In the revised manuscript we added the following text to the discussion:

“To decrease this flexibility, we have now used two previously developed conformationspecific nanobodies (NbRoco1 and NbRoco2) to stabilize the protein in the GTP-state (Leemans et al., 2020), allowing us to solve its structure using cryo-EM (Figure 1). Recently, Nbs have successfully been used to obtain structural insights in the conformational states of a number of highly dynamic proteins (Uchański et al, 2020). These studies established that Nbs bind antigens primarily by conformational selection rather than by induced fit (Manglik et al., 2017; Smirnova et al.,2015). Since NbRoco1 and NbRoco2 were generated by immunization with fully native CtRoco bound to a nonhydrolysable GTP analogue, and subsequently selected by phase display using the same functional protein, it is thus safe to assume that these Nbs bind to and stabilize a relevant conformation that is present within the “active” CtRoco conformational space (Leemans et al., 2020). Moreover, our current structures are also in very good agreement with previous biochemical studies and data from HDX-MS and negative stain EM (Deyaert et al., 2019; Deyaert, Wauters, et al., 2017).”

**Recommendations for the authors:**

**Reviewer #1:**
(1) Figure 2C: please label the residues with meshes (switch 2).

Labels have been added to figure 2C.

(2) A supplemental figure for the following statement will be helpful "A remarkable feature of the CtRoco dimer structure was the dimer-stabilized orientation of the P-loop, which would hamper direct nucleotide binding on the dimer. Correspondingly, in the current structure, the P-loop changes orientation, allowing GTPgS to bind, although the EM map does not allow unambiguous placement of the entire P-loop. Surprisingly, also the Switch 1 loop could not be fully modeled, which could indicate some flexibility in this region despite the presence of a GTP analog".

An additional Figure 2–figure supplement 2 has been added to illustrate this.

(3) A supplemental figure for the following statement will be helpful "A final important observation in the Roc domain concerns the very C-terminal part of Switch 2 (residues 520 to 533), which could not be modeled in our GTP bound structure due to flexibility, while in the nucleotide-free dimer structure this region is structured and located at the interface of the Roc domain with the LRR-Roc linker and CORA. In this way, the conformational changes induced by GTPgS binding could be relayed via the Switch 2 toward the LRR and CORA domains, and vice versa."

An additional Figure 2–figure supplement 2 has been added to illustrate this.

(4) A structural comparison of each domain (LRR, ROC, COR) between NF and GTP-bound states will be greatly useful to understand statements in the manuscript, such as "In addition to the Cterminal dimerization part of CORB that becomes unstructured, also other large conformational changes are observed in the CORA and CORB domains of CtRoco upon GTPgS binding."

We would like to clarify that with this statement we refer to changes in the relative orientation of the domains between the nucleotide-free and GTPgS-bound states, rather than to conformational changes within each domain. These changes in relative orientation are illustrated in Figure 2 and the associated Figure supplements.

(5) The statement "to a lesser extent, also between CDR1 and the LRR-Roc linker" is not clearlyillustrated in Figure 3B.

The reviewer is correct, and we now also show CDR1 in Figure 3B.

(6) Extra panels can be added in Figure 1 Sup. 4 to illustrate the following statement "In the density map NbRoco2 can easily be identified and placed on the concave side of the LRR domain... Nterminal and C-terminal b-strands interacting with the very C-terminal repeat of the LRR".

We belief the density map corresponding to NbRoco2 is clearly shown in Figure 1 – supplement 4A. A reference to this figure panel is now added to the main text.

(7) "In the presence of both Nbs, the hydrolysis rate was increased 4-fold compared to CtRocoL487A alone and 2-fold compared to CtRoco-L487A in the presence of NbRoco1 only, again illustrating a collaboration between the Nbs (Figure 5C)" Here, is it 6-fold instead of 4-fold?

The reviewer is correct. We changed this accordingly in the manuscript.

**Reviewer #2:**
(1) At many places in the manuscript the lack of structural details is explained by the assumed local flexibility of the protein. This may be true for many cases (such as linker regions), but is probably not always correct; several other explanations are possible to get no local structural details.

See our answer to point 2, below.

(2) At several other places in the manuscript the high flexibility is used to explain the lack of structural details (so the reasoning is reversed compared to point 1); this would require that a priori it is known that that the region is flexible and therefore no structure can be expected. An example is found mid-page 8: "A final important observation in the Roc domain concerns the very C-terminal part of Switch 2 (residues 520 to 533), which could not be modeled in our GTP bound structure due to flexibility, while in the nucleotide-free dimer structure this region is structured and located at the interface of the Roc domain with the LRR-Roc linker and CORA." As written there must be a reference to experiments showing the "due to flexibility"

The reviewer is correct that additional factors might affect the interpretability of the map, such as the small size of the regions used for the focused refinements (around 50 kDa each) or a preferential distribution of orientation of the particles in the grid. Particle distribution plots are now shown in Figure 1 – Figure supplements 1 and 2. However, due to the intrinsic flexible nature of the Switch 1 and Switch 2 regions, we assume this flexibility to be the major cause of lack of features in the EM maps, especially since some of the neighboring regions display well-resolved maps.

Nevertheless, in the manuscript we reworded our statements to be more careful. For example, on page 8:

“Also the Switch 1 loop could not be fully modeled in our structure, presumably indicating some flexibility in this region despite the presence of a GTP analogue.”

“… potentially due to flexibility of this region in the new position of the Switch 2…”